# Kinetic Alterations in Resurgent Sodium Currents of Mutant Na_v_1.4 Channel in Two Patients Affected by Paramyotonia Congenita

**DOI:** 10.3390/biology11040613

**Published:** 2022-04-18

**Authors:** Ming-Jen Lee, Pi-Chen Lin, Ming-Hong Lin, Hsin-Ying Clair Chiou, Kai Wang, Chiung-Wei Huang

**Affiliations:** 1Department of Neurology, National Taiwan University Hospital, Taipei 100, Taiwan; mjlee@ntu.edu.tw; 2Department of Medical Genetics, National Taiwan University Hospital, Taipei 100, Taiwan; 3Division of Endocrinology and Metabolism, Department of Internal Medicine, Kaohsiung Medical University Hospital, Kaohsiung 80756, Taiwan; pichli@kmu.edu.tw; 4Department of Microbiology and Immunology, School of Medicine, College of Medicine, Kaohsiung Medical University, Kaohsiung 807, Taiwan; mhlin@kmu.edu.tw; 5Center of Teaching and Research, Kaohsiung Municipal Siaogang Hospital, Kaohsiung Medical University, Kaohsiung 807, Taiwan; phoenixchiou@gmail.com; 6Kaohsiung Medical University Hospital, Kaohsiung Medical University, Kaohsiung 807, Taiwan; 7Institute for Systems Biology, Seattle, WA 98109, USA; kwang@systemsbiology.org; 8Department of Post Baccalaureate Medicine, Kaohsiung Medical University, Kaohsiung 807, Taiwan; 9Department of Physiology, Kaohsiung Medical University, Kaohsiung 807, Taiwan

**Keywords:** paramyotonia congenita, Na_v_1.4 channel, resurgent currents

## Abstract

**Simple Summary:**

Paramyotonia congenita (PMC) is a rare muscle disorder that causes myotonia and weakness of facial and limb muscles. The electromyography in PMC shows continual spontaneous, high-frequency spike potentials in skeletal muscles. Genetic mutations in the Na_v_1.4 channel that cause hyperexcitability of muscle fibers are responsible for PMC. However, the genotype–phenotype relationship is highly diversified, and the molecular pathology remains unclear. Here, we investigated the electrophysiology in the Na_v_1.4 channel with mutations, p.V781I and p.A1737T, which were found in two Taiwanese patients. We identified the distinct changes in gating mechanisms altered by mutations which may underlie the clinical phenotype.

**Abstract:**

Paramyotonia congenita (PMC) is a rare skeletal muscle disorder characterized by muscle stiffness upon repetitive exercise and cold exposure. PMC was reported to be caused by dominant mutations in the SCN4A gene encoding the α subunit of the Na_v_1.4 channel. Recently, we identified two missense mutations of the SCN4A gene, p.V781I and p.A1737T, in two PMC families. To evaluate the changes in electrophysiological properties caused by the mutations, both mutant and wild-type (WT) SCN4A genes were expressed in CHO-K1 and HEK-293T cells. Then, whole-cell patch-clamp recording was employed to study the altered gating of mutant channels. The activation curve of transient current showed a hyperpolarizing shift in both mutant Na_v_1.4 channels as compared to the WT channel, whereas there was a depolarizing shift in the fast inactivation curve. These changes confer to an increase in window current in the mutant channels. Further investigations demonstrated that the mutated channel proteins generate significantly larger resurgent currents as compared to the WT channel and take longer to attain the peak of resurgent current than the WT channel. In conclusion, the current study demonstrates that p.V781I and p.A1737T mutations in the Na_v_1.4 channel increase both the sustained and the resurgent Na^+^ current, leading to membrane hyperexcitability with a lower firing threshold, which may influence the clinical phenotype.

## 1. Introduction

Myotonia is a condition characterized by hyperexcitable muscles, manifested as a prolonged muscle stiffness after voluntary contractions and as a “myotonic run” of the skeletal muscle membrane in electromyography (EMG) records. [1,2,3,4]. Non-dystrophic myotonia with atypical increasing excitability is the primary clinical presentation of paramyotonia congenita (PMC) [1,2,3,4]. Patients with PMC may have attacks of weakness and muscle stiffness which are exacerbated by repetitive exercise and cold temperature [1,2,3,4]. PMC is associated with mutations in the SCN4A gene [1,2,3,4], which encodes the α-subunit of the Na_v_1.4 voltage-gated sodium channel protein. Apart from PMC, SCN4A mutations are also involved in the sodium channel myotonias that are not exacerbated by exercise or influenced by potassium [1].

By activating the Na_v_1.4 channel, the depolarization propagates to the T-tubule with the subsequent release of calcium ions leading to the contraction of skeletal muscle [5,6,7]. The pore-forming α-subunit contains four homologous domains (DI-DIV), each with six transmembrane segments (S1–S6), and it can bind one or more of the β-subunits to form a heteromultimeric protein complex [8,9]. The S4 segment contains positively charged amino-acid residues which act as a voltage sensor; upon depolarization, they can move outward and alter the channel conformation and function.

It has been reported that, in PMC, the enhanced inward Na^+^ current is due to impaired fast inactivation or enhancement of activation through Nav1.4 [5]. Moreover, there is an increase in excitability with a slight Na^+^ influx that leads to the initial burst of myotonia discharges and results in stiffness [5]. With increasing levels of discharge, most Nav1.4 channels will switch to the activated state and hold back the immediate re-elicitation, which can result in paralysis. [5]. Several known mutations of SCN4A associated with PMC, including p.Q270K (located in DI/S5–6 segments), p.N440K (in DI/S6), p.G1306E, p.T1313A, N1366S, and p.R1448P (all in DII-IV linker), are also known to result in the impairment of inactivation process and a slowing of the macroscopic inactivation kinetics of the channel [10,11,12,13,14,15,16]. Other mutations associated with PMCs, such as p.I693L in DII/S2, p.T1313A in the DIII–IV linker, p.G1456E in DIV/S4, p.R1448C in DIV/S4, p.R1448H in DIV/S4, and p.A1589M in DIV/S5, can only be seen in sodium channelopathies [7,10,11,17,18,19,20,21,22,23,24]. Despite our growing understanding of the molecular pathology associated with PMC, the changes in resurgent and sustained Na^+^ currents associated with the mutations and their correlation with disease severity are not fully explained.

Analyzing samples from two families with PMC in Taiwan, we identified two missense mutations in the SCN4A gene: p.V781I, located in the DII/S6, and p.A1737T, located in the C-terminus. Using whole-cell patch recordings of the cells expressing the mutant SCN4A cDNA clones, we conducted an electrophysiological analysis of the mutated channels. The results showed that these mutations altered the gating properties of the channel, leading to increased sustained and window currents during depolarization and increased resurgent currents during repolarization. The increased Na^+^ currents caused by the mutated channels, the likely root of hyperexcitability in PMC, could be attributed to the destabilization of inactivated states caused by an accelerated transition from inactivated to open Na_v_1.4 states. Moreover, enhanced resurgent current in mutated channels may also reduce the firing threshold. This study contributes to the understanding of the molecular basis of the PMC in the clinic.

## 2. Materials and Methods

### 2.1. Patients and the Mutations

The index cases were two individuals that visited the outpatient clinic in National Taiwan University Hospital, Taipei, Taiwan. One of the patients was a 21-year-old male who suffered from muscle stiffness with difficulties in relaxation of the limbs and eyelids. He developed weakness in both hands accompanied by tightness at finger extension. The symptoms were aggravated after repeat exercise and cold exposure. Painful muscle spasms occurred occasionally. His father and aunt also had similar symptoms since youth. The electromyography (EMG) study showed early recruitment with frequent myotonic discharges and a dive-bomber sound at the sampled muscles. Under the impression of paramyotonia congenita, sequencing of the coding regions of the *SCN4A* gene in the patient found a missense variant, c.2341G > A, which results in the p.V781I change. 

The second index case was a 32 year old male who suffered from muscle tightness and stiffness during running since childhood. The symptoms got worse on cold days or after repeated exercise. Percussion myotonia was observed on thumb adduction and wrist flexion. The repetitive stimulation test after long exercise showed reduced compound motor action potentials by stimulation of the right ulnar nerve. The EMG study revealed a short run of myotonic discharge in the right abductor pollicis brevis and biceps muscles. There was no family history. The clinical features fulfilled the diagnostic criteria of paramyotonia congenita. The SCN4A gene mutation screen identified a sequence variant, c.5209G > A, which results in the p.A1737T change. 

The study was approved by the Institutional Review Board of National Taiwan University Hospital, Taipei, Taiwan (201802049RINB). 

### 2.2. Expression and Molecular Biology of Na_v_1.4 Constructs

The human SCN4A gene was cloned into a pTracer-EF/V5-His vector. The transfected cells express a green fluorescent protein (GFP) driven by a separate promoter in the pTracer, making them identifiable under a fluorescent microscope (Nikon Inc., Tokyo, Japan). The clones with the c.2341G > A (p.V781I) and c.5209G > A (p.A1737T) mutations were created using the QuikChange Site-Directed Mutagenesis System Kit (Stratagene, La Jolla, CA, USA), and the sequences were verified by sequencing (3730xl DNA Analyzer; Applied Biosystems, Foster, CA, USA).

### 2.3. Preparation of Cell Lines for Transfection

Chinese hamster ovary (CHO-K1) and human embryonic kidney 293T (HEK-293T) cells were obtained from the Food Industry Research and Development Institute (Hsinchu, Taiwan). Protocols using these cell lines were approved by the Institutional Biosafety and Use Committee (IBUC) of Kaohsiung Medical University College of Medicine. The CHO-K1 cells were cultured in F12-K medium (Thermo Fisher Scientific, Waltham, MA, USA), and the HEK293T cells were cultured in Gibco Dulbecco’s Modified Eagle Medium (DMEM), both supplemented with 10% fetal bovine serum (FBS) and 1% penicillin–streptomycin, under standard conditions (37 °C, 5% CO_2_). The cells were then maintained until the electrophysiological recordings (within 3 days following DNA transfection). To prepare them for recordings, cells were first dissociated in their medium (F12-K for CHO-K1 and DMEM for HEK293T), supplemented with 10% FBS. Next, the cells were plated on glass coverslips (kept at 37 °C and 5% CO_2_). Lipofectamine^TM^ 3000 (Thermo Fisher Scientific) was used to transiently introduce the WT and mutant SCN4A cDNA clones. After 24 h, the cells were washed with the medium and incubated for 96 h. Before electrophysiological experiments, the cells were treated with 1.0 mg/mL protease type XXIII (Sigma Chemical Co., St Lois, MO, USA) for 15 min. 

### 2.4. Electrophysiological Recordings

Whole-cell patch-clamp recordings from CHO-K1 and HEK293T cells, expressing WT or p.V781I or p.A1737T channels, were performed at room temperature using the Multiple Clamps 700B multichannel amplifier (Axon Instruments, Sunnyvale, CA, USA) with pClamp 9.2 software (Molecular Devices, San Jose, CA, USA). Data were filtered at 10 kHz and digitalized at 50 kHz using a data acquisition interface (Digidata 1322, Axon Instruments). The pipettes were custom made from borosilicate glass tubes (Warner Instruments, MA, USA) using a horizontal puller (Zeitz Instruments, Inc. Martinsried, Planegg, Germany), and the tips were fire-polished (Narishige scientific instruments, Inc., Tokyo, Japan). The pipette resistance was 1–3 MΩ, filled with intracellular solutions containing 75 mM CsCl, 75 mM CsF, 5 mM HEPES, 2 mM CaCl_2_, and 2.5 mM EGTA, titrated to pH 7.4 with 1 M CsOH [20,21,22,23,24]. Whole-cell configuration was obtained with the formation of a giga-ohm seal in extracellular solution containing 150 mM NaCl, 2 mM MgCl_2_, 2 mM CaCl_2_, and 10 mM HEPES, titrated by 1 M NaOH to pH 7.4. The whole-cell patch-clamp cell was removed from the cover glass and placed in a linear array of pipes containing the extracellular solutions. We assessed resurgent currents by subtracting the leak currents and capacities produced by tetrodotoxin (TTX, Torcris Cookson, Langford, UK) [25,26,27]. The subunits of Na^+^ channels may be associated with several proteins, including the four Na_v_β subunits. The β1–4 subunits are inserted on the cell membrane with a transmembrane domain. Although the Na_v_β4 subunit contains 228 amino acids, only a short peptide from subunit β4 with approximately 14 residues was used in electrophysiological recording (KKLITFILKKTREK, the cytoplasmic tail of the entire β4 subunit) [27]. This peptide was dissolved in distilled water to make a 10 mM stock solution and then diluted into an intracellular solution to a final concentration of 0.1 mM [28,29].

### 2.5. Construction of Activation and Inactivation Curves

Patched cells were held at −120 mV for 100 ms, and then immediately subjected to a short step of test voltage, in the range of −140 to +40 mV in 5 mV increments. The peak inward current was plotted as a function of test voltage to generate the I–V plot. The Na^+^ currents usually reach maximum at a test voltage between 0 and −20 mV, and they decrease for more positive test voltages. We fitted a regression line to the linear part of the I–V curve (between +10 mV and +40 mV) for each cell, and we determined the maximal Na^+^ conductance (G_max_) by the slope, and the reversal potential (V_rev_) of Na^+^ currents by the intercept of the extrapolated line with the horizontal axis (i.e., at zero current). Then, the normalized sodium conductance plots, G/G_max_ versus V, were fitted with the Boltzmann function G/G_max_ = 1/(1 + exp(−(V*h* − V)/*k*)), determined by two parameters, V*h*, the potential at which activation is half-maximal, and *k*, the slope. We used this fit to determine the activation curve for each cell separately [30]. 

To generate the steady-state fast inactivation curve, we measured the current sweeps in response to a brief +10 mV test pulse after a 100 ms prepulse to a voltage ranging −140 to +40 mV in 5 mV increments from a holding potential of −120 mV. The peak currents, normalized by the maximum current in the prepulse series (I/I_max_), were plotted as a function of prepulse voltage and fitted with a Boltzmann function I/I_max_ = 1/(1 + exp((V − V*h*)/*k*)), where parameters V*h* (midpoint) and *k* (slope) are defined as for the inactivation curve [30].

### 2.6. Data Analysis and Statistics

Patch-clamp data were analyzed using Clampfit 9.2 (Axon Instrument, Sunnyvale, CA, USA). The V*h* and *k* of the steady-state activation curves were determined using Boltzmann functions and choosing the best fit one for the curve. To determine the recovery time constants of the resurgent current, responses of each cell were fitted to a first-order exponential function, and the time constants derived were averaged from cells as indicated. Statistical analysis was done using the statistical software Sigmaplot version 10.0 (Systat Software Inc.,wpcubed GmbH, Erkrath, Germany). Student’s independent *t*-test and ANOVA followed by two-tailed Bonferroni’s correction were used to test the differences between groups. Statistical significance was defined as *p* < 0.05.

## 3. Results

### 3.1. The p.V781I and p.A1737T Mutant Channels Showed Larger Sustained Na^+^ Currents Than the WT Na_v_1.4 Channel

To elucidate the functional consequences of the SCN4A mutations identified from the PMC patients, the electrophysiological characteristics of the WT, p.V781I, and p.A1737T mutant channels were compared using whole-cells patch-clamp recordings performed on cells transiently expressing the protein. The changes in the activation and inactivation process of the three Na_v_1.4 channels are shown in Figure 1 and Figure 2. In Figure 1A, the sample sweeps from these three channels were demonstrated. The cells were held at −120 mV and then subjected to the influx of depolarizing current for 150 ms, escalated from −140 to +40 mV stepwise. Both mutant channels exhibited a hyperpolarizing shift of the activation curve when compared with the WT channel (Figure 1B,C). The half-activation potential (V_1/2_) was significantly higher in the mutant channels as compared to the WT channels (Figure 1C). The slope of the activation curve did not show a significant change in the p.V781I and p.A1737T mutant channels (Figure 1D). To investigate whether the kinetics of inactivation were changed, we measured the time constants (tau) of fast inactivation (Figure 1E) in the channels by fitting the decaying phases of transient Na^+^ currents from 80% of maximal to the end with a mono-exponential function. The fast inactivation time constants for the p.V781I and p.A1737T mutant channels were significantly larger than for the WT channel between −40 and −20 mV. 

The electrophysiological changes in the inactivation process of the WT and two mutant Nav1.4 channels were also evaluated. The current traces shown in Figure 2A were the recordings from the cells expressing WT and mutant (p.V781I and p.A1737T) channels. After conditioning of depolarizing with incremental potentials from −140 to +40 mV, the inactivation curves were observed following a test pulse at +10 mV. The inactivation curves shifted toward depolarization (Figure 2B) and the inactivation potential V_1/2_ was significantly depolarized in cells with the mutant channels as compared to the WT (Figure 2C). The slopes of the activation and inactivation curves were similar between the mutants and the WT channels (Figure 1D and Figure 2D). 

The window current, defined by the area under the activation and inactivation curves, was used to measure the presumed sustained Na^+^ current. The comparison of window current between the WT and two mutant channels is shown in Figure 3A. The window currents from the p.V781I and p.A1737T mutant channels were significantly greater than the WT channel (Figure 3B). We calculated the product of ratios of conductance (G/G_max_) and steady-state inactivation (I/I_max_) from the observed activation and inactivation curves at different potentials (−20, −40, −60, and −80 mV) to determine the sustained Na^+^ currents at the specific potentials (Figure 3C). Productions in the mutant Na_v_1.4 channels were significantly greater than those of the WT channel (Figure 3C). The ratios of sustained current to peak current (Figure 3D) were significantly higher for the two mutant channels compared to the WT channel, indicating a meaningful increase in the sustained Na^+^ currents in the mutant channels. The hyperpolarizing shifts in the activation and depolarizing shifts in the inactivation curves increased the window or sustained Na^+^ currents in the mutant channels.

### 3.2. The Mutant Na_v_1.4 Channels Increase the Resurgent Na^+^ Currents

Resurgent Na^+^ currents can be observed in voltage-gated Na^+^ channels with an internal Navβ4 peptide during the repolarization process [29,31,32,33]. The traces of resurgent Na^+^ currents at the repolarization (0 to −120 mV) stage in the mutant channels (p.V781I and p.A1737T) and WT are shown in Figure 4A. The sweeps showed that the magnitudes of the resurgent Na^+^ currents in both mutant channels were significantly larger than in the WT channel. Compared to WT channels, the ratios of resurgent to transient currents were higher for the mutant channels at repolarizing voltages between −70 mV and −10 mV (Figure 4B).

The kinetic changes between activation and inactivation states are probably related to the genesis of resurgent Na^+^ currents. Our results suggest that both the p.V781I mutation, located at domain II/segment VI (DII/S6), and the p.A1737T mutation, located at the C-terminus, may be involved in the increase in resurgent current. The increase in resurgent currents in the mutant channels caused membrane hyperexcitability with a reduction in the firing threshold, which can cause a drastic increase in muscular stiffness and clinical symptoms of PMC.

### 3.3. There Are Probably Two Individual Open States for Transient and Resurgent Na^+^ Currents in the Na_v_1.4 Channel

According to our previous studies, the Nav1.4 channel may have two different open states which govern the genesis of transient and resurgent Na^+^ currents [29,33]. To evaluate the activation kinetics of both transient and resurgent currents, the cells were subjected to an incremental depolarization current (from −100 to +180 mV) for 50 ms, followed by repolarization at −60 mV for 150 ms. A depolarization pulse for 10 ms can cause most Na^+^ channels to enter an open state. The WT and the p.V781I and p.A1737T mutants channels showed the same activation curves of transient Na^+^ currents when depolarized for 10 ms (Figure 5A,B). The blue traces demonstrate that inactivated currents in the p.V781I and p.A1737T mutant channels were slower than those in the WT channels. Depolarization with a 10 ms prepulse opened most resurgent Na^+^ channels. A comparison of the resurgent and transient Na^+^ currents in Figure 5B revealed that the activation curves of resurgent Na^+^ currents were less voltage-dependent, given the less steep activation curves of resurgent current compared to those of transient current. Compared to the activation curves of transient Na^+^ currents, the curves of the resurgent currents showed a depolarizing shift (Figure 5B). Furthermore, the slopes of the activation curves (transient vs. resurgent) were different from each other, which may indicate two disparate open states of the sodium channel. While focusing on the activation curve of resurgent current, the p.V781I mutant channels exhibited a hyperpolarizing shift, even at low repolarization voltages, compared to the WT channel (Figure 5B). The voltage of V_1/2_ in the p.V781I mutant channel was significantly smaller than in the WT channel (Figure 5C), whereas the slope of the activation curve was significantly milder in p.V781I mutant channels. In addition, the slope of the activation curve in the p.V781I mutant channel was greater than in the other two (WT and p.A1737T), indicating that this mutant channel is also involved in voltage sensor movement (Figure 5D). In contrast, the p.A1737T mutant channel exhibited a somewhat depolarization shift relative to the WT channel (Figure 5B). These findings indicate that there may be two different open states for the activation of transient and resurgent current. The missense mutations in Na_v_1.4 result in biophysical changes with increasing resurgent current, leading to membrane hyperexcitability.

### 3.4. Resurgent Na^+^ Current Decay Rates and Time to Peak in the Mutant Channels

In addition to the magnitude of the resurgent Na^+^ currents, the kinetic changes caused by the mutant channels p.V781I and p.A1737T were also evaluated. Within a wide range of prepulse depolarization, from +40 to +180 mV (Figure 6A), the decay of resurgent Na^+^ current represented by the reciprocal value of the time constant (tau) was assessed. The values of 1/tau were not significantly altered in the p.V781I and p.A1737T mutant channels as compared to the WT channel. We also assessed the time needed to generate the resurgent current (time to peak, in Figure 6B) at repolarization (−60 mV) of these channels. The time to peak with variable prepulses of depolarization (between +40 and +180 mV) were significantly longer in mutant channels (p.V781I and p.A1737T) than in the WT channel (Figure 6B).

In addition, the changes in these kinetic parameters with a series of different repolarizing potentials from −70 to −20 mV (Figure 7A) were also evaluated in both mutant and WT Na_v_1.4 channels. The WT channel exhibited a significantly shorter time to peak during repolarization than the p.V781I and p.A1737T mutant channels (Figure 7A). The decay kinetics (1/tau) of the p.V781I and p.A1737T mutant channels matched well with the WT channel (1/tau(v) = 0.2 × exp(−0.6 V/25) ms^−1^ for the WT channel, 0.1 × exp(−0.9 V/25) ms^−1^ for the p.V781I mutant, and 0.1 × exp(−0.8 V/25) ms^−1^ for the p.A1737T mutant) (Figure 7B). These findings suggest that the decay rates of the resurgent Na^+^ currents in p.V781I and p.A1737T mutant channels were similar to that of the WT channel. The generation of resurgent Na^+^ currents in the mutant channels was noticeably slower than in the WT channel, indicating that the mutation causes a longer open state, either after strong depolarizing prepulses (Figure 6B) or under different repolarization potentials (Figure 7A).

### 3.5. The Tail Currents in the WT and the p.V781I and p.A1737T Mutant Channels Were Unchanged

Similar to the resurgent current, the tail current in the voltage gated sodium channel took place after depolarization. Herein, we also evaluated the tail current, as well as the change in its decay kinetics, among the WT, p.V781I, and p.A1737T channels. To observe the tail current, cells were subjected to a repolarization current for 20 ms after a short prepulse of depolarization (+40 mV for 0.5 ms, Figure 8A). The voltages for repolarization ranged from −80 mV to −20 mV (Figure 8A). Current traces in Figure 8A showed that the tail currents of the mutant channels were seemingly retained longer than those of the WT channel. While considering the decay kinetics, there were no significant changes in the deactivation rates, represented by 1/tau, between WT and mutant channels. The difference between them was less than 0.4 ms^−1^ (Figure 8A,B). Furthermore, there was a difference in the decay kinetics between the resurgent currents and the tail currents (Figure 7B vs. Figure 8B), indicating two independent biophysical properties. These findings suggest that there may be two different open states responsible for transient and resurgent Na^+^ currents.

## 4. Discussion

### 4.1. Biophysical Changes Resulting from Two Mutant Nav1.4 Channel Proteins in Paramyotonia Congenita

The study evaluated the biophysical changes resulting from two mutated Nav1.4 channels. Electrophysiological data from whole-cell patch-clamp recording were collected from cells expressing the WT SCN4A gene and SCN4A gene carries missense mutations, p.V781I or p.A1737T. These two mutations were identified from two independent patients with paramyotonia congenita. The minor allele frequencies of the p.V781I (rs62070884, c.2341G > A) among different populations are 2.4% (American), 2.2% (east Asian), 0.6% (European), 0.1% (south Asian), and 0% (African) (https://asia.ensembl.org/Homo_sapiens/Variation/Population?db=core;g=ENSG00000007314;r=17:63938554-63972918;t=ENST00000435607;v=rs62070884;vdb=variation;vf=108011886,Ensemblrelease106, accessed on 13 April 2022). Although the frequency is relatively high in American and east Asian populations, the clinical conditions of the patients with this variant include paramyotonia congenita, potassium-aggravated myotonia, hypokalemic periodic paralysis type 2, familial hyperkalemic periodic paralysis, congenital myasthenic syndrome, and acetazolamide-responsive myotonia, etc., (accession number, VCV000021153, https://www.ncbi.nlm.nih.gov/clinvar, accessed on 16 December 2021). Therefore, the authentication for the pathogenicity of this sequence variant remains elusive. The missense mutation, p.A1737T (rs758381788, c.5209G > A), is located at the C-terminal of the α-subunit of Na_v_1.4 channel. The minor allele frequency is null among the populations tested (https://asia.ensembl.org/Homosapiens/Variation/Population?db=core;g=ENSG00000007314;r=17:63938554-63972918;t=ENST00000435607;v=rs758381788;vdb=variation;vf=108011886,Ensemblrelease106, accessed on 13 April 2022). It is a de novo mutation in the affected family. To explore the pathogenicity of these missense mutations, we investigated their impacts on the biophysical property of the Nav1.4 channel. 

To assess any correlation between genotype and clinical presentation, the clinical details and electrophysiological data were reviewed. The age of onset for the patient with p.V781I mutation is at childhood (~6 years of age), whereas it was at adolescence in the patient with p.A1737T mutation. Tightness and weakness of the eyelid were found in patients with p.V781I. The severity of muscle stiffness was mild, and the level of creatine kinase was 267 U/L (normal range, <220 U/L) in the patient with p.A1737T mutation. Although the extent and the time to peak of resurgent current are more severe in the p.A1737T mutant, the correlation between the biophysical indication and clinical severity remains obscure. A few biophysical investigations from the mutated SCN4A gene have been reported [7,10,11,17,18,19,20,21,22,23,24]; however, a positive correlation between the functional consequences and genotype has not been documented. It might be due to factors, such as modifier genes, genetic backgrounds, and environmental or metabolic conditions.

### 4.2. Proposal of a New Open State Responsible for Resurgent Na^+^ Current 

To summarize the result in this and previous studies on the sodium channels [29,31,32,33], we conceived a new model of the gating mechanism that adds a few new internal states and transitions to the old model. The new model has two distinct open states (O1 and O2) that correspond to the two fast inactivated states (I1 and I2). This can help to explain the genesis of transient and resurgent Na^+^ currents in the human Nav1.4 channel. According to our previous model, the Na_v_β4 peptide may bind to the channel, implicating changes in its gating states [29,31,32,33]. There might be a change in binding energy when interacting with the internal Navβ4 peptides.

Resurgent currents are only discernible in the Na_v_1.4 channel when Na_v_β4 peptide is present (Figure 4). The conventional model of the resurgent currents is based on the inactivating peptide competes with Na_v_β4 peptide for blocking of the channel pore openings [34,35,36,37]. However, our previous study showed that the addition of Na_v_β4 peptide leads to significantly larger resurgent currents in the Na_v_1.7 mutant than in the WT channel [31]. There are a few disparate points against the conventional resurgent currents model. Firstly, at negative voltages, the time constant for decay kinetics of resurgent current is faster than tail currents (Figure 6 and Figure 7), which suggests there might be different gating states for these two independent currents. Furthermore, the difference in the activation curve between the resurgent and the transient currents also indicates that there may be two distinct open states (Figure 5) [29,33]. Secondly, for a moderately depolarizing prepulse, the resurgent current decay with a 10–20 ms time constant should be shorter in the competition model. Thirdly, in the presence of Na_v_β4 peptide, the activation curve of the resurgent current shows a substantially depolarizing shift [29,31,32,33]. As a competition model, there may be a hyperpolarizing shift in the presence of Na_v_β4, rather than a depolarizing shift. Therefore, we proposed that the Na_v_β4 peptide may act as a gating modifier, inducing a gating conformation change, rather than an inactivating peptide as a pore blocker. We anticipate that the mutations located at domain II (DII/S6, p.V781I) and the C-terminus (p.A1737T) of the Nav1.4 channel probably play a role in inactivating Na^+^ channels and/or regulating conduction pathways [30,38,39,40].

### 4.3. Sustained and Resurgent Na^+^ Currents Are Increased in the Mutant p.V781I and p.A1737T Channels

The two mutant Na_v_1.4 channels exhibited a marked increase in sustained and resurgent Na^+^ currents in a wide range of membrane voltages. The biophysical alterations caused by either the p.V781I or p.A1737T mutation could contribute to the enhancement of repetitive discharges with different firing patterns in muscles. The increase in sustained and resurgent currents may lead to a rapid transition from an inactivated state to an active one (a “destabilized” inactivated state). The hyperpolarizing shift in the activation curve and depolarizing shift in inactivation caused by the mutations increased the sustained current (Figure 1B, Figure 2B and Figure 5B), evidencing the uncoupling of activation and inactivation. These findings provide insights into the molecular pathogenesis of PMC. The p.V781 located at DII/S6 may interacts with the S4–S5 linker, which is close to the internal pore mouth serving as the structural element. The voltage sensor movements with conformational changes in the activation and inactivation gates play a critical role in activation/inactivation coupling [41,42,43,44,45]. Our results are consistent with the notion that the conformational changes caused by the mutations p.V781I and p.A1737T contribute to the impairment of inactivation, as well as the uncoupling of activation and inactivation. 

## 5. Conclusions

Mutations of the Na_v_1.4 channel result in myotonic myopathies, such as myotonic congenita, periodic paralysis, and PMC, presenting with a wide range of clinical symptoms [46]. The study demonstrated the increased sustained and window currents, as well as a larger magnitude of resurgent currents with a longer time to peak in the mutant (p.V781I and p.A1737T) Na_v_1.4 channels. These findings may contribute to the escalating membrane potential, lowering the firing threshold rendering muscle hyperexcitability, which is the hallmark of myotonia.

## Figures and Tables

**Figure 1 biology-11-00613-f001:**
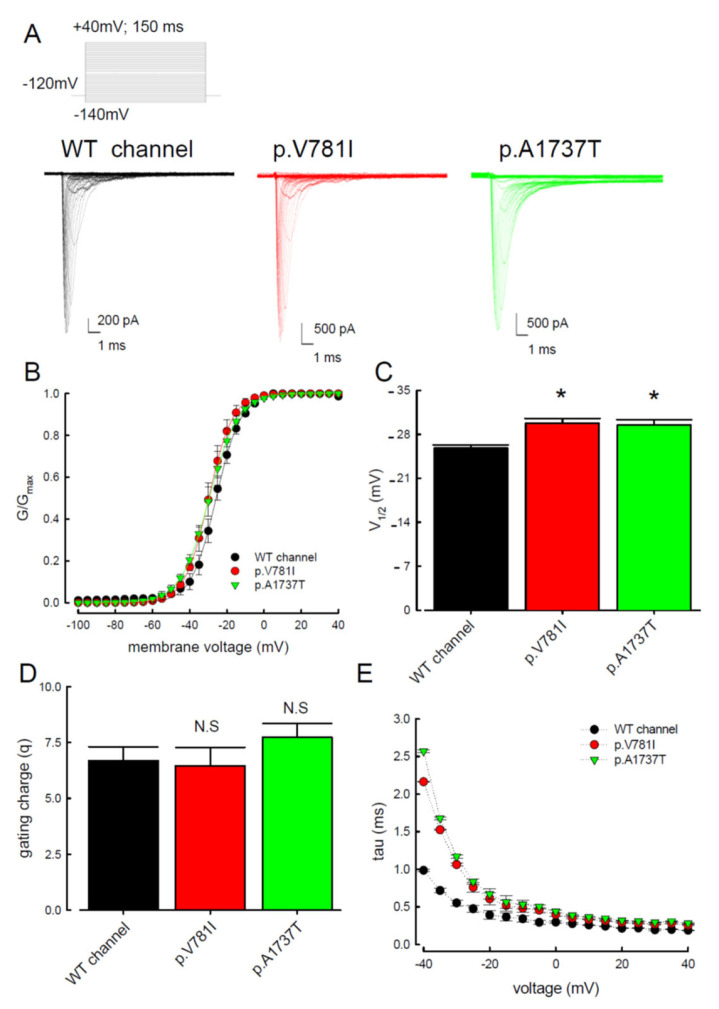
Analysis of the WT and the p.V781I and p.A1737T mutant sodium current activation process. (**A**) Cells voltage-clamped at −120 mV were subjected to a series of 150 ms long pulses, sampled between −140 and +40 mV in 5 mV increments. Sample sweeps from WT, p.V781I, and p.A1737T mutant Nav1.4 channels were recorded. (**B**) Sodium current activation curves (continuous lines) fitted to the data (symbols) by the Boltzmann function. Data are expressed as means and standard deviations (*n* = 10). (**C**) The half-activation potential V_1/2_ of these different channels was −25.8 ± 0.2 mV, −29.8 ± 0.6 mV, and −29.5 ± 1.3 mV for the WT, p.V781I, and p.A1737T mutant channels, respectively. Data are expressed as means ± SEM (*n* = 10; * *p* < 0.05 compared to control). (**D**) Note that the cumulative data displayed indicate that the slope (*k*) for the Na_v_1.4 channel was associated with the WT, p.V781I, and p.A1737T mutant channels. Data are expressed as means ± SEM (*n* = 10, N.S. = no statistically significant difference compared to control). (**E**) The fast inactivation time constants (tau) were obtained by fitting the decay phase of transient Na^+^ currents of WT, p.V781I, and p.A1737T mutant channels shown in (**A**) with a mono-exponential function. Data are expressed as means ± SEM.

**Figure 2 biology-11-00613-f002:**
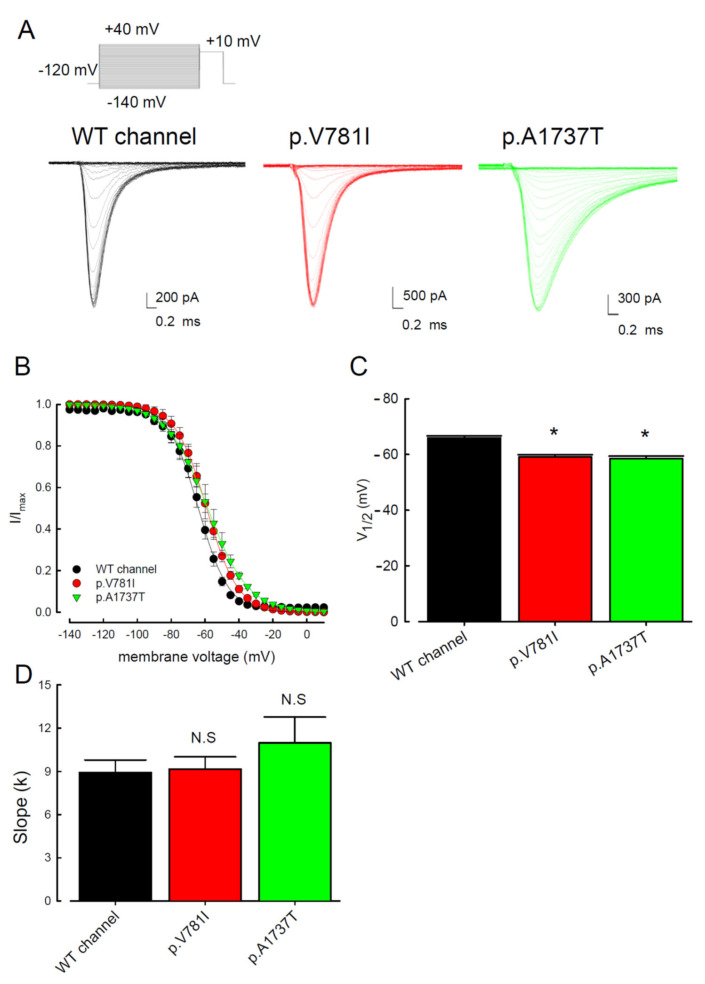
Analysis of the WT and the p.V781I and p.A1737T mutant sodium current inactivation process. (**A**) Current traces were recorded in the WT and the mutant p.V781I and p.A1737T Nav1.4 channels in response to a brief +10 mV test pulse preceded by a conditioning pulse in a two-pulse voltage protocol (see Section 2 for more details). (**B**) Voltage dependence of the quasi-steady-state inactivation. Peak sodium currents in response to the +10 mV test pulse are plotted as a function of the conditioning potential, evoked in a 5 mV stepwise escalating from −140 to +40 mV (*n* = 10). (**C**) The half-inactivation voltage, V_1/2_, of Nav1.4 channels was approximately −66.0 ± 0.7 mV, −59.1 ± 0.8 mV, and −58.5 ± 0.9 mV for the WT, p.V781I, and p.A1737T channels, respectively (*n* = 10). (**D**) The cumulative data that slope (*k*) of the Nav1.4 channel for the WT, p.V781I, and p.A1737T mutant channels. Data are expressed as means ± SEM (* *p* < 0.05, *n* = 10; N.S., no statistically significant difference compared to control).

**Figure 3 biology-11-00613-f003:**
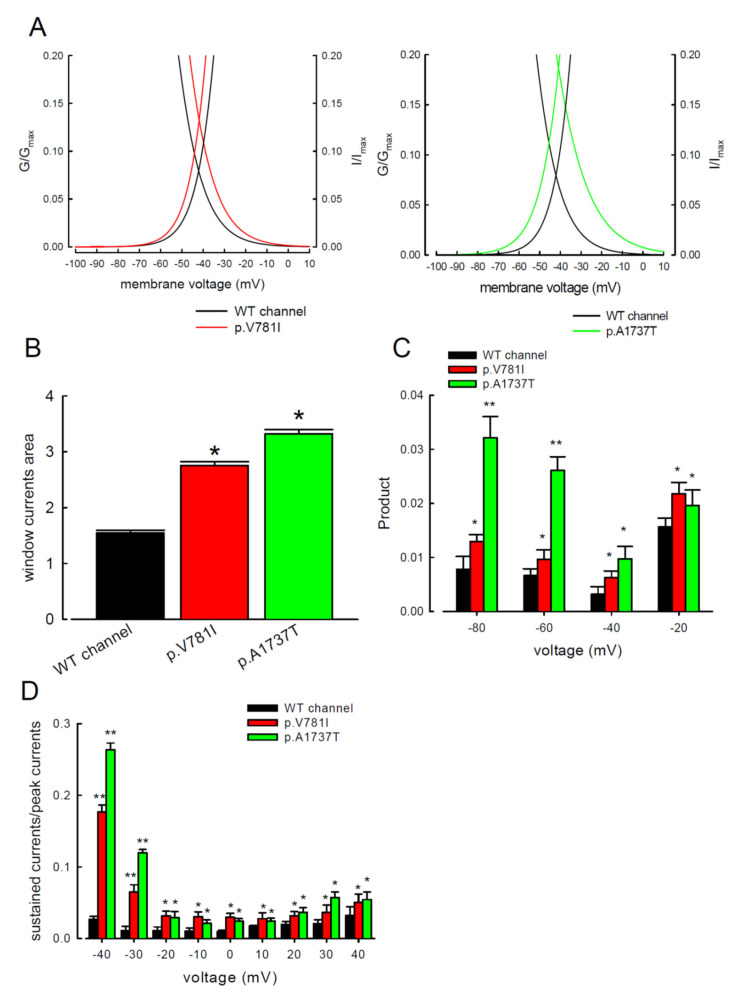
The p.V781I and p.A1737T mutant channels had extensive window currents. (**A**) The black (WT) and colored (p.V781I, red; p.A1737T, green) lines are the zoomed-in portions of the activation and inactivation curves in Figure 1A and Figure 2A. The window currents were calculated in the range between −100 and +10 mV. (**B**) The area of the window current, which was under the intersection of the activation curve (+10mV) and the inactivation curve (−100 mV), was calculated using Python 9.0 [16]. The vertical bar graph shows the integral calculation of the area under activation and inactivation curves of the three channels (* *p* < 0.05). (**C**) The values of the product of the ratios G/G_max_ and I/I_max_ (activation from Figure 1A multiplied by inactivation from Figure 2A) are plotted against the specific voltage at −80, −60, −40, and −20 mV in the WT, p.V781I, and p.A1737T mutant channels (* *p* < 0.05 and ** *p* <0.01). (**D**) The ratio between the sustained (currents averaged between 90 and 100 ms of the pulse) and the peak transient Na^+^ currents was significantly higher in the p.V781I and p.A1737T mutant channels than in the WT channel (* *p* < 0.05, and ** *p* < 0.01).

**Figure 4 biology-11-00613-f004:**
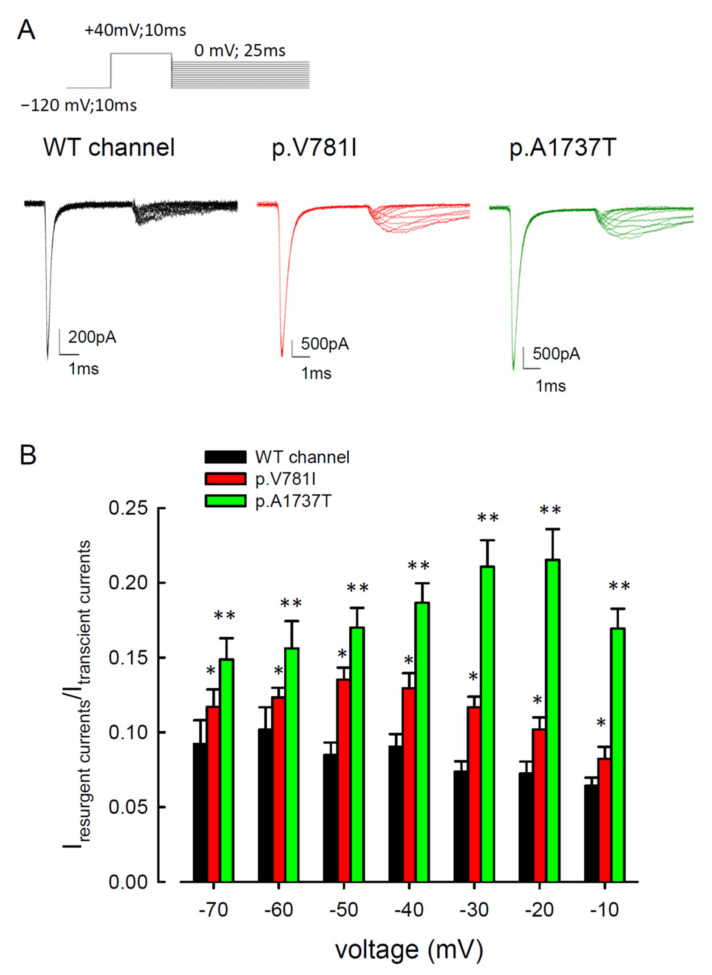
There were larger resurgent Na^+^ currents in the p.V781I and p.A1737T channels than in the WT channel. (**A**) In the presence of 0.1 mM Na_v_β4 peptide in the extracellular medium, cells were patched at −120 mV, and the resurgent Na^+^ currents were measured at the repolarization stage by applying a stepwise voltage escalating from 0 and −120 mV in 10 mV increments, following a +40 mV depolarizing prepulse. (**B**) The population results (*n* = 8) were obtained in the experiment in part **A**. There was a significant difference in the resurgent to peak transient current ratios (I resurgent/I transient) between the WT and the p.V781I and p.A1737T mutant channels at all repolarization potentials, between −70 and −10 mV (* *p* < 0.05 and ** *p* < 0.01).

**Figure 5 biology-11-00613-f005:**
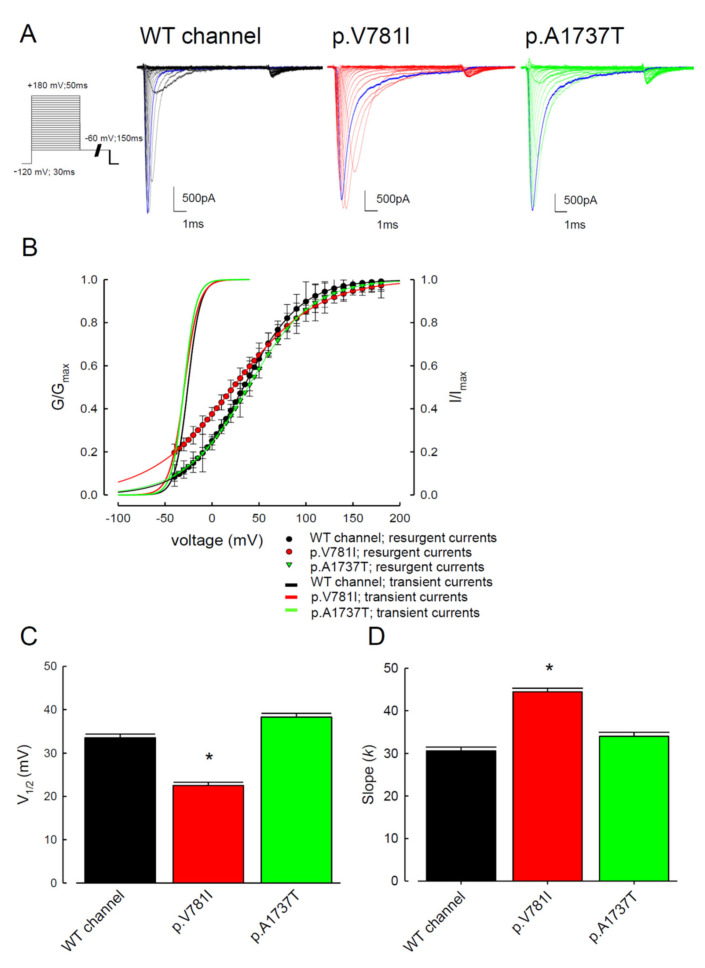
The steady-state activation curve for resurgent Na^+^ currents in the WT and the p.V781I and p.A1737T mutant channels. (**A**) The cells were held at −120 mV for approximately 30 ms, and then subjected to various 10 ms depolarizing prepulse sampling between −100 and +180 mV range with stepwise 10 mV increments for about 50 ms. Repolarization pulses evoked the resurgent Na^+^ currents at −60 mV for approximately 150 ms. We obtained sample sweeps from the WT and the p.V781I and p.A1737T mutant Na_v_1.4 channels. The blue traces demonstrate that inactivated currents in the p.V781I and p.A1737T mutant channels were slower than those in the WT channels. (**B**) The steady-state activation curves of the transient (fitted lines in Figure 1B replotted for comparison) and resurgent Na^+^ currents. The resurgent Na^+^ activation curve for each cell was obtained by fitting the data with a Boltzmann function. (**C**,**D**) The population data (*n* = 8) for V*h* and *q*. The mean values were 33.4 ± 0.8 mV and 30.6 ± 0.8 for the WT channel, respectively. Those for the p.V781I mutant channel were 22.5 ± 0.7 mV (*n* = 8; * *p* < 0.05) and 44.4 ± 0.9 (*n* = 8; * *p* < 0.05), and those for the p.A1737T mutant channel were 38.27 ± 0.8 mV (*n* = 8; * *p* < 0.05) and 33.9 ± 0.9.

**Figure 6 biology-11-00613-f006:**
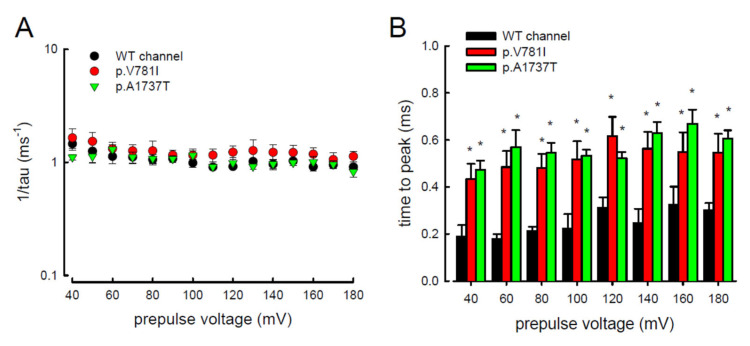
The kinetics of decay and the time for activation of resurgent Na^+^ currents in the WT, p.V781I, and p.A1737T channels. (**A**) The reciprocal of decay time constants (1/tau) of the resurgent Na^+^ currents obtained at −60 mV after the depolarizing prepulse between +40 and +180 mV in 20 mV increments for approximately 10 ms in the WT and the p.V781I, and p.A1737T mutant channels (*n* = 8). (**B**) With different prepulse voltages, the time to peak to generate resurgent Na^+^ current was faster in the WT channel than in the p.V781I and p.A1737T mutant channels. (*n* = 8 *; *p* < 0.05).

**Figure 7 biology-11-00613-f007:**
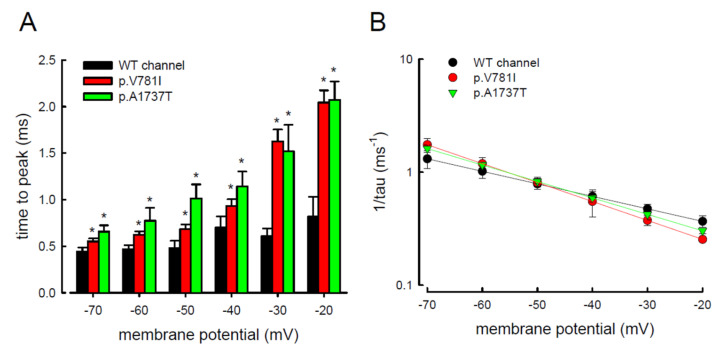
Slower time to peak of the resurgent Na^+^ currents in the p.V781I and p.A1737T mutant channels. (**A**) The time to peak in resurgent Na^+^ current was measured using the same protocol as in Figure 4. The value of time to peak plotted against the repolarization potentials for the WT and the p.V781I and p.A1737T mutant channels. The time to peak for the resurgent Na^+^ current was significantly shorter in the WT channel than in the p.V781I and p.A1737T mutant channels, at variable repolarization membrane potentials between −20 and −70 mV (* *p* < 0.05). (**B**) The reciprocal time constants (1/tau) for the decay phase of resurgent Na^+^ currents were plotted against the repolarizing potentials in semilogarithmic scales for the WT and the p.V781I and p.A1737T mutant channels. The lines were linear regression fitted using the formulas 1/tau_(V)_ = 0.2 × exp(−0.6 V/25) ms^−1^, 0.1 × exp(−0.9 V/25) ms^−1^, and 0.1 × exp(−0.8 V/25) ms^−1^ for the WT and the p.V781I and p.A1737T mutant channels, respectively. “V” is the membrane potential in mV.

**Figure 8 biology-11-00613-f008:**
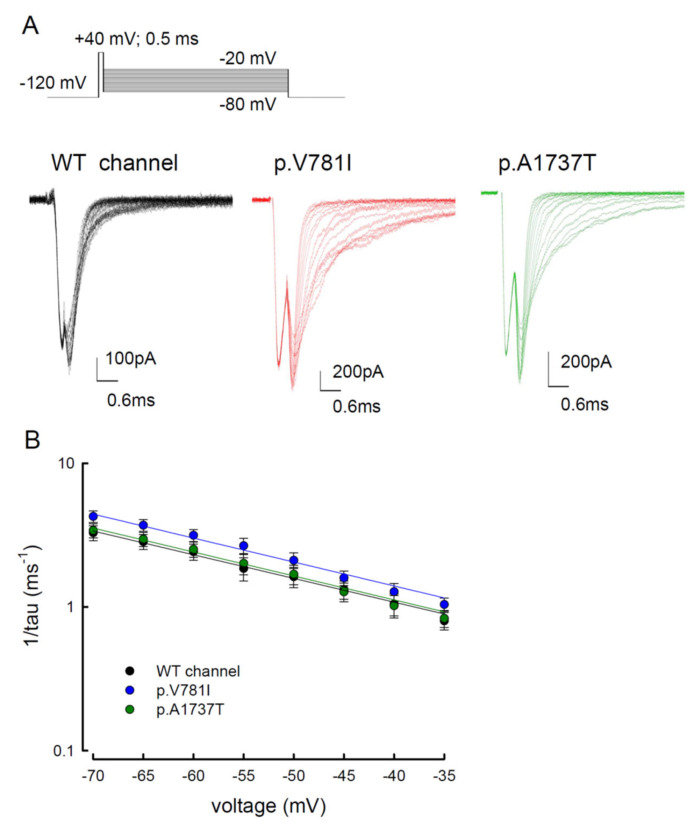
The tail current protocol was used to estimate the steady-state activation curve for the WT, p.V781I, and p.A1737T mutant channels. (**A**) The cells were held at −120 mV and depolarized with a voltage of +40 mV for approximately 0.5 ms of the activation pulse, followed by repolarization from −80 mV to −20 mV for approximately 20 ms in the WT, p.V781I, and p.A1737T channels. The tail currents showed faster decay kinetics as the deactivating pulse became more negative. (**B**) The decay phase of tail currents in (**A**) was fitted using a mono-exponential function for different deactivating potentials in the WT, p.V781I, and p.A1737T channels. The reciprocal of time constants of the decaying phase in tail currents was plotted against voltages in a semi-logarithmic scale for the WT, p.V781I, and p.A1737T channels. The lines were linear regression fitted using the formulas 1/tau_(V)_ = 0.2 × exp(−0.9 V/25) ms^−1^, 0.2 × exp(−0.9 V/25) ms^−1^, and 0.2 × exp(−0.9 V/25) ms^−1^ for the WT, p.V781I, and p.A1737T channels, respectively.

## Data Availability

Not applicable.

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
