# Peer review of "Kinetic Alterations in Resurgent Sodium Currents of Mutant Nav1.4 Channel in Two Patients Affected by Paramyotonia Congenita"

_biology, 2022, doi:10.3390/biology11040613_

Round 1

Reviewer 1 Report

see attached

Author Response

Response to Reviewer#1 Comments

Reviewer #1

We thank the reviewer for very helpful advice on the manuscript. We have carefully revised the manuscript with responses one-by-one successively as follows.

Major comments:

[1] b-subunits of Nav channels used could be unclear. How can these subunits be attached to a-subunits of the channels, since they should insert and attach to the transmembrane portion? In lines 138-144, please elaborate it clearly.

RESPONSE:

We apologize for the suboptimal expression in the previous manuscript. We are rewrite the sentence in the lines 160-166.

We assessed resurgent currents by subtracting the leak currents and capacities produced by tetrodotoxin (TTX, Torcris Cookson, Langford, UK) [1-3]. The subunits of Na+ channels may be associated with several proteins, including the four Navb subunits. The b1-4 subunits are inserted on the cell membrane with a trans-membrane domain. Although Navb4 subunit contains 228 amino acids, only a short peptide from the subunit with approximately 14 residues were used in electrophysiological recording (KKLITFILKKTREK, the cytoplasmic tail of the entire b4 subunit) [3].

[2] In line 155, the term in the equation should be changed to “….exp(-(Vh-V)/k)”

RESPONSE:

Many thanks for the suggestion! We have changed the equation “1/[1+exp(-(Vh-V)/k)]” in the revised manuscript [line 179].

[3] In line 156, the Boltzmann function has broadly been used. References 25-32 could be unnecessarily quoted. One could be adequately stated at most.

RESPONSE:

We followed the suggestion and only cited only the references [4] [lines 181 and 187].

[4] In lines 167-168, the sentence of “….. from the best fitting …” needs to be rephrased.

RESPONSE:

We have revised the sentence: “The midpoint (Vh) and slope (k) of the steady-state activation curves were determined using Boltzmann functions and choosing the best fit one for the curve”.

[5] In lines 172-173, the analyses should be included with ANOVA-one way or ANOVA-two way.

RESPONSE:

Based on the reviewer’s comments, we revised the sentence into “Students' independent t-tests and ANOVA followed by two-tailed Bonferroni's corrections were used to test the differences between groups. Statistical significance was defined as p < 0.05” [lines 190-197 in the revised manuscript].

[6] In lines 220-221, “fast” might be removed, since steady-state inactivation could be sufficiently described. More strictly, quasi-steady-state inactivation curve of the curve had better be used.

RESPONSE:

According to the reviewer’s suggestion, we have changed into “qusi-steady state inactivation” in lines 252 in the revise manuscript.

[7] “sampled” needs to be changed to “evoked”.

RESPONSE:

We changed “sampled” to “evoked” in line 253.

[8] In lines 229-231 (or line 156) and in Figure 2D, it is highly recommended to use the value of gating charge (q) instead of k for the evaluation on the Boltzmann function. It would be hence recommended to estimate changes in free energy for the activation of resurgent sodium current in these missense mutants (i.e., p.V781I and p.A1737T). In particular, in line 432, it is speculated that change in binding energy with internal Navb4 peptides might occur.

RESPONSE:

In Figure 2D, the “slope (k)” has been changed into “gating charge (q)” in the line 257, and added to rewrite the sentence “There might be a change in binding energy while binding with the internal Navb4 peptides” in the line 480.

[9] In line 235, “quantitation” needs to be used with caution. How was the quantitative measure of window Na+ current conduced, instead of graphical representation? Please estimate the area of window Na+ current in WT channels and pV781I or p.A1737T mutant channels.

RESPONSE:

Many thanks for the comments! Measuring the area under the activation and inactivation curve to represent window current as shown in our figure 3, A and B, has been proposed by Farinato et al. [5]. To measure the area, the Python software has been employed for the integral calculation. We added the sentence “The integral calculation of area of window current which was under the intersection of the activation curve (+10mV) and the inactivation curve (-100mV), has been performed by Python 9.0 software” [lines 280-282].

[10] In Figure 4A (middle portion), please check the time-to-peak of resurgent current. The value in p.V781I mutant channels appears to be raised. Likewise, the duration of time-to-peak in transient inward Na+ currents in response to abrupt depolarizing pulse was noticeably changed. Please provide the expanded current traces in Figure 4A for better illustration.

RESPONSE:

We apologize for the suboptimal raw sweeps in the WT, p.V178I, and p.A1737T channels. To have a better illustration, we replotted the raw sweep in Figure 4A and actually, the time-to-peak in transient inward currents seems to be protracted in both mutant channels as compared to the WT channel (lower panel of Figure 4A).

[11] In Figure 5A, the inactivation properties of current traces in p.V781I and p.A1737T mutant channels have been noticeably slowed. The change in the activation curve of resurgent currents could have been seriously disturbed by the inactivation process of the current. Again, it is recommended for the authors to make representative traces thicker for clear illustrations.

RESPONSE:

As suggested, we have made the raw sweeps thicker for clear illustration of the inactivation current trace (raw sweep 4) in Figure 5A. The blue traces illustrated that the inactivated currents in p.V781I and p.A1737T mutant channels are slower than the WT channel [lines 324-326, and 351-352, in the revised manuscript].

[12] In lines 268-269, the sentence needs to be rephrased. “inter-transformation” is unclear and hence needs to be elaborated clearly. The authors need to discuss this issue in detail.

RESPONSE:

Many thanks for the suggestion! We speculate that since the window current increase in mutant channels resulting in elevation of membrane potential, the threshold for activation of resurgent current would be decreased at repolarization state. Nevertheless, the assumption has not been validated. We revised the sentences into, “The kinetic changes between activation and inactivation states are probably related to the genesis of resurgent Na+ currents. Our results suggest that both the p.V781I mutation, located at domain II/segment VI (DII/S6), and the p.A1737T mutation, located at the C-terminus, may involve in this mechanism” [lines 300-303].

[13] Most of representative current traces are quit “thin” and hence become unclear. Please use thick line for each trace for better presentation.

RESPONSE:

As suggested, we have thickened the raw sweeps in Figures 1, 2, 4, and 5.

[14] In Figure 5A, current traces are unclear. The line is quite thin and hence barely seen. It is better to have the expanded traces.

RESPONSE:

As suggested, the raw sweeps in Figures 5A has been redrew.

[15] In lines 366-368, please show representative traces of how the reciprocal time constants were derived.

RESPONSE:

We fitted the decay phase of the resurgent currents from the 95% of the peak to the steady-state current by a standard exponential function: f(x)=A×exp(-t/t)+C.

Figure 1. The reciprocal time constants of resurgent currents are shown in p.V781I mutant channel, sweep number 9.  The blue line represents a fitted exponential function.

Minor comments:

[1] In line 32 and line 416, alpha is better to be changed with “a”.

RESPONSE:

We have change to “a” in lines 33 and 452 in the revised manuscript.

[2] In lines 39-42, the sentence shown herein is irrelevant to the study and better to be rephrased or removed.

RESPONSE:

Many thanks for the suggestion! We rephrased these sentences into, “Further investigations demonstrated that the mutant channels generate a significant large resurgent current as compared to the WT channel. There was a longer time to attain the peak of resurgent current in the mutant channels than the WT channel.”.

[3] In lines 61-63, the sentence needs to be rephrased.

RESPONSE:

We have rephrased the sentence to, "By activating the Nav1.4 channel, the depolarization propagates to T-tubule with subsequent release of calcium ion leading to the contraction of skeletal muscle” in the revised manuscript [line 60-61].

[4] In lines 120-121, the sentence is unclear. Does it mean that the cells which were unable to be transfected (i.e., cells with wild type) were NOT made?

RESPONSE:

Sorry for the unclear explanation. In this article, the data is presented with CHO-K1 cells" [lines 139-141]. Nevertheless, HEK293T cells have also been transfected and the data from the mutant expressed cells was consistent with what we found in CHO-K1 cells.

[5] In line 134-135, the composition of CaCl2 needs to be removed to be affected by endogenous voltage-gated Ca2+ currents. “pH7.4” needs to be changed to “pH 7.4”.

RESPONSE:

There is voltage gated calcium channel on the membrane of CHO-K1 cells. Nevertheless, the influence of CaCl2 and the calcium current is not the target for our investigation. Such an influence is probably not a major determinant for us to investigate the comparison of sodium current between mutant and WT sodium channels.

As suggested, the pH7.4 has been changed to pH 7.4.

[6] In lines 135-137, the sentence is unclear and should be rephrased. For example, what does “pipette-attached cell” mean?

RESPONSE:

We apologized the error present in previous manuscript, and we replace the “pipette-attached cell” to the “whole cell-patch clamp cell” in the revise manuscript [line 158].

[7] In line 162, “sampled” needs to be replaced with “ranging”.

RESPONSE:

As suggested, we replaced the “sample” to “ranging” in line 183 of the revised manuscript.

[8] In lines 193-194, the sentence is unclear. How were current responses confirmed?

RESPONSE:

The sentence “Current responses confirmed the whole-cell patch-clamp configuration” has been removed.

[9] The formula in the equation can be removed, since it is illustrated in Materials and Methods section.

RESPONSE:

Thanks for reviewer’s commends. We remove the formula in the revised manuscript.

[10] In lines 223-224 and lines 227-228, the sentence can be removed, since the statement has been described in Materials and Methods.

RESPONSE:

The sentences have been removed in the revised manuscript.

[11] In line 276, the title needs to be changed to “……. than those in WT channel.”

In line 277, “0.1mM” should be changed to “0.1 mM”.

RESPONSE:

We have changed the title and sentence to “…than those in WT channel”, and to “0.1 mM”, respectively [line 307 and 308].

[12] In line 272, “results” needs to be changed to “result”.

RESPONSE:

We have changed the typo [line 304].

[13] In line 288, “Na+” needs to be changed to “Na+”.

RESPONSE:

As suggested, we have changed it to “Na+” [line 319].

[14] In lines 294-295, the sentence is unclear and the authors hence needs to rephrase it.

RESPONSE:

We revised the sentenceDepolarization with a 10-ms prepulse will open most resurgent Na+ channels. A comparison of the resurgent and transient Na+ currents in Figure 5B, revealed that the activation curves of resurgent Na+ currents were less voltage-dependent, given the less steep of the activation curves of resurgent current compare to those of transient current.” [lines 327-330].

[15] In line 460, “gate” needs to be changed to “gating”. In Figure 9, it would be better to have theoretical results for the good matching with experimental observations. If not, appropriate reference(s) with respect to such theoretical model should be quoted in the manuscript.

RESPONSE:

Thanks for the reviewer`s comments; however, reviewer#2 strongly suggested that we need to remove Figure 9 and rewrite the DISCUSSION. We removed figure 9 and legend 9, and rewrote the DISCUSSION in the revised manuscript.

[16] In line 525, “consistent” needs to be replaced with “are consistent”.

RESPONSE:

We have changed the word to “are consistent” in line 515.

[17] In line 533, “escalating membrane potential” is confusing. “membrane depolarization” could be better. Hence, please provide the values of input resistance and resting membrane potential in the examined cells.

RESPONSE:

As suggested, the revised manuscripts changed the term to, "membrane depolarization". We are sorry to say that we cannot provide the values of input resistance and resting membrane potential in transfected CHO-K1 and HEK-293T cells.

[18] The representative data in the text of the manuscript could be adequately used as the values with one decimal digit. It is not necessary to use two decimal digit appearing throughout the text of the manuscript.

RESPONSE:

Thank you for the reviewer's comments. Representative data in the revised manuscript are expressed as values with one decimal digit.

[19] No direct experimental observations in muscle hyperexcitability was provided in this manuscript. The title needs to be changed to some extent to “Possible kinetic alterations in resurgent sodium currents of mutant Nav1.4 channel”

RESPONSE:

We agree with the reviewer's comments. The title has been changed into, "The possible kinetic alterations in resurgent sodium currents of mutant Nav1.4 channel resulting in paramyotonia congenita" in the revised manuscript.

[20] There are several grammatical and typo errors or flaws which are necessarily corrected in the manuscript. To have a significant proofread for the manuscript is highly recommended.

RESPONSE:

Many thanks for the comments. This manuscript has review by a native English speaker to improve readability. Certification is presented as the following sheet.

Reference

  1. Khaliq, Z.M.; Gouwens, N.W.; Raman, I.M. The contribution of resurgent sodium current to high-frequency firing in Purkinje neurons: an experimental and modeling study. J Neurosci 2003, 23, 4899-4912.
  2. Grieco, T.M.; Raman, I.M. Production of resurgent current in NaV1.6-null Purkinje neurons by slowing sodium channel inactivation with beta-pompilidotoxin. J Neurosci 2004, 24, 35-42, doi:10.1523/JNEUROSCI.3807-03.2004.
  3. Grieco, T.M.; Malhotra, J.D.; Chen, C.; Isom, L.L.; Raman, I.M. Open-channel block by the cytoplasmic tail of sodium channel beta4 as a mechanism for resurgent sodium current. Neuron 2005, 45, 233-244, doi:10.1016/j.neuron.2004.12.035.
  4. Yang, Y.C.; Hsieh, J.Y.; Kuo, C.C. The external pore loop interacts with S6 and S3-S4 linker in domain 4 to assume an essential role in gating control and anticonvulsant action in the Na(+) channel. J Gen Physiol 2009, 134, 95-113, doi:jgp.200810158 [pii]10.1085/jgp.200810158.
  5. Farinato, A.; Altamura, C.; Imbrici, P.; Maggi, L.; Bernasconi, P.; Mantegazza, R.; Pasquali, L.; Siciliano, G.; Lo Monaco, M.; Vial, C., et al. Pharmacogenetics of myotonic hNav1.4 sodium channel variants situated near the fast inactivation gate. Pharmacological research 2019, 141, 224-235, doi:10.1016/j.phrs.2019.01.004.

Reviewer 2 Report

The paper by Lee and collaborators is interesting. They functionally characterized two Nav1.4 mutations found in patients suffering from Paramyotonia Congenita.

The study is generally well performed.

However, the manuscript must be improved in many parts. The methods section requires more details. Conclusions should be mitigated by the limitation of the study. Much attention should be paid to references. English language requires revision.

Importantly, the authors already proposed the schemes for sodium channel gating in previous studies (see for instance, Huang et al., PlosBiol 2016). This could be interpreted as duplication of data and auto-plagiarism. The authors should deeply revise the discussion.

Specific comments:

Simple summary, line 26: the statement “the molecular pathology remains elusive” is a little bit strong. Many mutations have been already functionally characterized and molecular mechanisms have been described.  

Abstract, line 31:  “by a dominant mutation” to be substituted by “by dominant mutations”

Introduction

Line 64: The exact number of auxiliary β-subunits bound to the α-subunit is not well defined. It would be fairer to indicate heteromultimeric with the binding of one or more β-subunits.

Lines 50 to 74. It would be fairer citing review articles. Actually, references 1 and 6 deal with CLCN1 mutations that are out the topic of the Ms. References 2 to 7 are specific experimental studies that are little suitable for general concepts. Examples of recent relevant reviews are:

Cannon SC. Sodium Channelopathies of Skeletal Muscle. Handb Exp Pharmacol. 2018;246:309-330.  (already cited ref. 45)

Maggi L, Bonanno S, Altamura C, Desaphy JF. Ion Channel Gene Mutations Causing Skeletal Muscle Disorders: Pathomechanisms and Opportunities for Therapy. Cells. 2021 Jun 16;10(6):1521.

Stunnenberg BC, LoRusso S, Arnold WD, Barohn RJ, Cannon SC, Fontaine B, Griggs RC, Hanna MG, Matthews E, Meola G, Sansone VA, Trivedi JR, van Engelen BGM, Vicart S, Statland JM. Guidelines on clinical presentation and management of nondystrophic myotonias. Muscle Nerve. 2020 Oct;62(4):430-444. 

Desaphy JF, Altamura C, Vicart S, Fontaine B. Targeted Therapies for Skeletal Muscle Ion Channelopathies: Systematic Review and Steps Towards Precision Medicine. J Neuromuscul Dis. 2021;8(3):357-381.

Line 70: more than “leading to”, enhanced inward currents are “due to”

Line 75-76: There are errors in the mutations and references are not correct: p.G1306E is not a PMC mutation; It has been associated to sodium channel myotonia, myotonia permanens and SNEL subtypes. Mutation p.G270L does not exist; the correct name should be Q270K (Carle et al., 2009).

Line 78. References 4, 10-12 reported Q1633E, R1448P, N440K, and T1313A mutations. References for G1306E, N1366S, and Q270K are missing.

Line 81: Again, references do not match the listed mutations. The mutation p.A1589D does not exist.

Line 85: Mutation p.V781I is not novel. See Miller et al., Neurology 2004; Stunnenberg et al., Neuromusc Disord 2018. As stated in the discussion, it also appears in public databases.

Methods

What was the rationale for using two different types of cell lines? In the results, there is no more indication of the cells used for each kind of experiment. More information is needed.

Line 105 : method for transfection is actually not described.

Lines 137-141: unclear, please revise English.

Line 152: with no sodium ions in the intracellular patch solution, what would be the Vrev value of Na+ currents? This is quite unusual using a sodium-free pipette solution. Please give more information and justification.

Results:

More information regarding the patients would be appreciated. How the diagnosis of paramyotonia congenita was reached?

Line 211: V1/2 is not smaller; it is shifted toward less negative values.

Line 232: the window current has been correlated with mutation severity (see for instance Farinato et al., Pharmacol res 2019)

Figures 1A and 1B, please note that the kinetics of entry into fast inactivation appear slower for A1737T compared to WT and V781I. Such an effect is usually reported as a key mechanism for myotonia. Further analysis of current decay kinetics would be interesting.

Figure 4A and 5A: please indicate the duration of the voltage steps on the protocole.

Discussion

There is no attempt to correlate the biophysical defects in nav1.4 with the severity or any specific feature of the phenotype.

As said above, the two schemes have been already published by the authors.

One important issue is that the use of the Navβ4 peptide is far from being physiological. It is not sure that such resurgent currents may occur in skeletal muscle fibers. This limitation of the study should be disclosed in the discussion.

Discussing the paper of Jarecki et al (JCI 2010) would be important as it shows resurgent currents induced by a PMC mutation expressed in neuronal cells.

Author Response

Response to Reviewer# 2 Comments

We thank the reviewer for very helpful advice on the manuscript. We have carefully revised the manuscript.

Specific comments:

Simple summary, line 26: the statement “the molecular pathology remains elusive” is a little bit strong. Many mutations have been already functionally characterized and molecular mechanisms have been described.

RESPONSE:

Thanks for the reviewer's comments. We changed the word from "elusive" to "unclear " in the revised manuscript [line 27].

Abstract, line 31: “by a dominant mutation” to be substituted by “by dominant mutations”

RESPONSE:

As suggested, we changed the "a dominant mutation" to "by dominant mutations" in the revised manuscript [line 32].

Introduction

Line 64: The exact number of auxiliary β-subunits bound to the α-subunit is not well defined. It would be fairer to indicate heteromultimeric with the binding of one or more β-subunits.

RESPONSE:

We changed the sentence to “The a-subunit of Nav1.4 can bind one or more of β-subunits to develop a heteromultimeric molecule. The pore-forming a-subunit, contains four homologous domains (DI-DIV), each with six transmembrane segments (S1~S6)” [lines 62-64].

Lines 50 to 74. It would be fairer citing review articles. Actually, references 1 and 6 deal with CLCN1 mutations that are out the topic of the Ms. References 2 to 7 are specific experimental studies that are little suitable for general concepts. Examples of recent relevant reviews are:

  1. Cannon SC. Sodium Channelopathies of Skeletal Muscle. Handb Exp Pharmacol. 2018;246:309-330. (already cited ref. 45)
  2. Maggi L, Bonanno S, Altamura C, Desaphy JF. Ion Channel Gene Mutations Causing Skeletal Muscle Disorders: Pathomechanisms and Opportunities for Therapy. Cells. 2021 Jun 16;10(6):1521.
  3. Stunnenberg BC, LoRusso S, Arnold WD, Barohn RJ, Cannon SC, Fontaine B, Griggs RC, Hanna MG, Matthews E, Meola G, Sansone VA, Trivedi JR, van Engelen BGM, Vicart S, Statland JM. Guidelines on clinical presentation and management of nondystrophic myotonias. Muscle Nerve. 2020 Oct;62(4):430-444.
  4. Desaphy JF, Altamura C, Vicart S, Fontaine B. Targeted Therapies for Skeletal Muscle Ion Channelopathies: Systematic Review and Steps Towards Precision Medicine. J Neuromuscul Dis. 2021;8(3):357-381.

RESPONSE:

Thank you for the reviewer’s suggestions. We have changed the reference citations.

Line 70: more than “leading to”, enhanced inward currents are “due to”

RESPONSE:

As suggested, we changed the phrase to “due to” in the revised manuscript [line 68].

Line 75-76: There are errors in the mutations and references are not correct: p.G1306E is not a PMC mutation; It has been associated to sodium channel myotonia, myotonia permanens and SNEL subtypes. Mutation p.G270L does not exist; the correct name should be Q270K (Carle et al., 2009).

RESPONSE:

Many thanks for the critical reading and suggestion. We correct the mutation to “p.Q270K” (Carle et al., 2009) in the revised manuscript.

Line 78. References 4, 10-12 reported Q1633E, R1448P, N440K, and T1313A mutations. References for G1306E, N1366S, and Q270K are missing.

RESPONSE:

As suggested, we have further cited a few references, the G1306E (Farinato et al., 2019), N1366S (Ke et al., 2017), and Q270K (Carle et al., 2009) in the revised manuscript.

Line 81: Again, references do not match the listed mutations. The mutation p.A1589D does not exist.

RESPONSE:

As suggested, we have corrected the mutation and cited the reference in the revised manuscript.

Line 85: Mutation p.V781I is not novel. See Miller et al., Neurology 2004; Stunnenberg et al., Neuromusc Disord 2018. As stated in the discussion, it also appears in public databases.

RESPONSE:

As suggested, we leave out the word, “novel” [line 82].

Methods

What was the rationale for using two different types of cell lines? In the results, there is no more indication of the cells used for each kind of experiment. More information is needed.

RESPONSE:

To avoid the false positive or negative results due to the cell-type specific effects, we carried out the experiments in two different cell lines. The results from the two types of cells line (CHO-K1 and HEK-293T) are similar. Thus, we present the results from what we obtained from CHO-K1 cells.

Line 105: method for transfection is actually not described.

RESPONSE:

I have apologized for the missing. We have added the following sentence, “LipofectamineTM 3000 (Thermo Fisher Scientific) was used for transient transfection using the WT and mutant cDNA clones of SCN4A gene. After 24 hours of transfection, the cells were washed with the medium” in the revised manuscript [lines 139-141].

Lines 137-141: unclear, please revise English.

RESPONSE:

We thank reviewer’s very important suggestions. We revised the sentences into “We assessed resurgent currents by subtracting the leak currents and capacities produced by tetrodotoxin (TTX, Torcris Cookson, Langford, UK) [1-3]. The subunits of Na+ channels may be associated with several proteins, including the four Navb subunits. The b1-4 subunits are inserted on the cell membrane with a trans-membrane domain. Although Navb4 subunit contains 228 amino acids, only a short peptide from the subunit with approximately 14 residues were used in electrophysiological recording (KKLITFILKKTREK, the cytoplasmic tail of the entire b4 subunit) [3]”in the revised manuscript.

Line 152: with no sodium ions in the intracellular patch solution, what would be the Vrev value of Na+ currents? This is quite unusual using a sodium-free pipette solution. Please give more information and justification.

RESPONSE:

Many thanks for the suggestion! There have been many studies on the human Nav1.4 channel or other voltage–gated Na+ channels, including those carried out in native (e.g., DRG) neurons and in cell lines (e.g., CHO–K1 and HEK293–T cells). In most studies, replacement of intracellular Na+ ions with Cs+ (to reduce the conductance through different “background” channels) [2,4-19] and use of the regression line on the rising arm of the current–voltage plot (e.g., +20 to +40 mV) for the prediction of full conductance at more negative voltages are very common practice [2,4-19]. Because Cs+ can also permeate through Na+ channels, the measurements of reversal potential would not be a problem. However, the regression line method should be sufficient or even better (considering the possibility of conductance changes near the reversal potential) for construction of the activation curve. The activation data therefore should be quite reliable.

Results:

More information regarding the patients would be appreciated. How the diagnosis of paramyotonia congenita was reached?

RESPONSE:

Thanks to the reviewer’s suggestion! We have added two paragraphs in the section of “Material and methods” of the revised manuscript.

Line 211: V1/2 is not smaller; it is shifted toward less negative values.

RESPONSE:

Thanks to the reviewer’s suggestion, we have changed the word from “smaller” to “hyperpolarization” in the revised manuscript [line 245].

Line 232: the window current has been correlated with mutation severity (see for instance Farinato et al., Pharmacol res 2019)

RESPONSE:

As suggested, in the revised manuscript, we describe how to calculate the area to represent the window current and cite the reference [20].

Figures 1A and 1B, please note that the kinetics of entry into fast inactivation appear slower for A1737T compared to WT and V781I. Such an effect is usually reported as a key mechanism for myotonia. Further analysis of current decay kinetics would be interesting.

RESPONSE:

As suggested, we evaluate the kinetics of fast-inactivation in the WT and mutant channels. To investigate whether the kinetics of inactivation were changes by p.V781I, and p.A1737T mutant channels, we measure the time constants (tau) of fast-inactivation (Figure. 1E) in the channels. The time constants were measured by fitting the decaying phases of transient Na+ currents from 80% of maximal to the end with a mono-exponential function. The fast-inactivation time constants of p.V781I, and p.A1737T mutant channels was significantly larger than WT channel between -40 and -20 mV.

Figure 4A and 5A: please indicate the duration of the voltage steps on the protocol.

RESPONSE:

As suggested, we indicate the duration of each voltage step on the protocol in Figure 4A and 5A.

Discussion

There is no attempt to correlate the biophysical defects in Nav1.4 with the severity or any specific feature of the phenotype.

RESPONSE:

Many thanks for the suggestion! In the revised manuscript, the following paragraph has been added into the section of “Discussion”.

To assess any correlation between mutation genotype and clinical severity, the clinical details and electrophysiological data have been reviewed. The age of onset for the patient with p.V781I mutation is at childhood (~ 6 years of age), but at adolescent in the patient with p.A1737T mutation. The tightness and weakness with involvement of eyelid was found in patients with p.V781I. The severity of muscle stiffness is mild and the level of creatine kinase is 267 U/L (normal range, < 220 U/L) in the patient with p.A1737T mutation. Although the extent and the time to peak of resurgent current is more severe in the p.A1737T mutant, correlation of between the biophysical manifestation and clinical severity remains obscure. A few biophysical investigations from the mutant SCN4A gene have been reported; however, a positive correlation between functional consequences and genotype has not been documented. It might be due to a few factors, such as modifier genes, genetic backgrounds and environmental or metabolic conditions.

As said above, the two schemes have been already published by the authors.

RESPONSE:

According to the reviewer’s very important comments, to avoid duplication of data and auto-plagiarism. We leave out the details about the explanation of the scheme which we proposed before.

One important issue is that the use of the Navβ4 peptide is far from being physiological. It is not sure that such resurgent currents may occur in skeletal muscle fibers. This limitation of the study should be disclosed in the discussion.

RESPONSE:

Many thanks for the comments. To test whether the resurgent current can be obtained in the skeletal muscles, the whole-cell patch-clamp recording has been performed on the C2C12 cells, a myogenic differentiated cell line. The resurgent Na+ currents can be observed in the myogenic differentiated cells from C2C12 cell lines containing a Navβ4 peptide at the repolarization stage. On the contrary, no resurgent Na+ currents can be found without containing the Navβ4 peptides. We will design to perform the experiments in skeletal muscle cells in the future.

Discussing the paper of Jarecki et al (JCI 2010) would be important as it shows resurgent currents induced by a PMC mutation expressed in neuronal cells.

RESPONSE:

Many thanks for the suggestion! We would added the following paragraph in the “Discussion section”.

The a subunits of Na+ channels may associate with several proteins on the cell membrane, including five b subunits [21-25]. The entire b4 subunit contains 228 amino acids, but the b4 peptide frequently used in the studies of resurgent Na+ currents contains only 14 residues (KKLITFILKKTREK, the cytoplasmic tail of the entire b4 subunit) [3]. Previous studies demonstrated that the full-length β4 subunit is well expressed in dorsal root ganglions (DRG) neuron. The expression level is especially high in large, but low in small and intermediate sensory neurons [21-23]. Nav1.1, Nav1.2, Nav1.3, and Nav1.6 channels are widely expressed in the central nervous system, whereas Nav1.4, channel is preferentially expressed on the human skeletal muscle [26]. After the TTX-R modification, co-expression of the b4 subunit and the alpha-subunits of these VGSCs, but not that of WT Nav1.4 in the dorsal root ganglion cells can generate resurgent current [27]. The expression of mutant Nav1.4 channel in the DRG neurons showed a significant increase of resurgent current [27]. Given the observed transient and resurgent currents in the Nav1.4 channel in this study, the disparate results may be due to the TTX-R modification or the different of expression cells (DRG v.s. CHO-K1).

Reference

  1. Khaliq, Z.M.; Gouwens, N.W.; Raman, I.M. The contribution of resurgent sodium current to high-frequency firing in Purkinje neurons: an experimental and modeling study. J Neurosci 2003, 23, 4899-4912.
  2. Grieco, T.M.; Raman, I.M. Production of resurgent current in NaV1.6-null Purkinje neurons by slowing sodium channel inactivation with beta-pompilidotoxin. J Neurosci 2004, 24, 35-42, doi:10.1523/JNEUROSCI.3807-03.2004.
  3. Grieco, T.M.; Malhotra, J.D.; Chen, C.; Isom, L.L.; Raman, I.M. Open-channel block by the cytoplasmic tail of sodium channel beta4 as a mechanism for resurgent sodium current. Neuron 2005, 45, 233-244, doi:10.1016/j.neuron.2004.12.035.
  4. Cheng, X.; Dib-Hajj, S.D.; Tyrrell, L.; Wright, D.A.; Fischer, T.Z.; Waxman, S.G. Mutations at opposite ends of the DIII/S4-S5 linker of sodium channel Na V 1.7 produce distinct pain disorders. Molecular pain 2010, 6, 24, doi:10.1186/1744-8069-6-24.
  5. Raman, I.M.; Sprunger, L.K.; Meisler, M.H.; Bean, B.P. Altered subthreshold sodium currents and disrupted firing patterns in Purkinje neurons of Scn8a mutant mice. Neuron 1997, 19, 881-891.
  6. Ahn, H.S.; Dib-Hajj, S.D.; Cox, J.J.; Tyrrell, L.; Elmslie, F.V.; Clarke, A.A.; Drenth, J.P.; Woods, C.G.; Waxman, S.G. A new Nav1.7 sodium channel mutation I234T in a child with severe pain. Eur J Pain 2010, 14, 944-950, doi:10.1016/j.ejpain.2010.03.007.
  7. Cheng, X.; Dib-Hajj, S.D.; Tyrrell, L.; Te Morsche, R.H.; Drenth, J.P.; Waxman, S.G. Deletion mutation of sodium channel Na(V)1.7 in inherited erythromelalgia: enhanced slow inactivation modulates dorsal root ganglion neuron hyperexcitability. Brain 2011, 134, 1972-1986, doi:10.1093/brain/awr143.
  8. Cheng, X.; Dib-Hajj, S.D.; Tyrrell, L.; Waxman, S.G. Mutation I136V alters electrophysiological properties of the Na(v)1.7 channel in a family with onset of erythromelalgia in the second decade. Molecular pain 2008, 4, 1, doi:10.1186/1744-8069-4-1.
  9. Choi, J.S.; Dib-Hajj, S.D.; Waxman, S.G. Inherited erythermalgia: limb pain from an S4 charge-neutral Na channelopathy. Neurology 2006, 67, 1563-1567, doi:10.1212/01.wnl.0000231514.33603.1e.
  10. Cummins, T.R.; Dib-Hajj, S.D.; Waxman, S.G. Electrophysiological properties of mutant Nav1.7 sodium channels in a painful inherited neuropathy. J Neurosci 2004, 24, 8232-8236, doi:10.1523/JNEUROSCI.2695-04.2004.
  11. Raman, I.M.; Bean, B.P. Resurgent sodium current and action potential formation in dissociated cerebellar Purkinje neurons. J Neurosci 1997, 17, 4517-4526.
  12. Kuo, C.C.; Bean, B.P. Slow binding of phenytoin to inactivated sodium channels in rat hippocampal neurons. Mol Pharmacol 1994, 46, 716-725.
  13. Yang, Y.C.; Kuo, C.C. Inhibition of Na(+) current by imipramine and related compounds: different binding kinetics as an inactivation stabilizer and as an open channel blocker. Mol Pharmacol 2002, 62, 1228-1237.
  14. Yang, Y.C.; Kuo, C.C. An inactivation stabilizer of the Na+ channel acts as an opportunistic pore blocker modulated by external Na+. J Gen Physiol 2005, 125, 465-481, doi:jgp.200409156 [pii]

10.1085/jgp.200409156.

  1. Huang, C.W.; Lai, H.J.; Huang, P.Y.; Lee, M.J.; Kuo, C.C. The Biophysical Basis Underlying Gating Changes in the p.V1316A Mutant Nav1.7 Channel and the Molecular Pathogenesis of Inherited Erythromelalgia. PLoS biology 2016, 14, e1002561, doi:10.1371/journal.pbio.1002561.
  2. Huang, C.W.; Lai, H.J.; Huang, P.Y.; Lee, M.J.; Kuo, C.C. Anomalous enhancement of resurgent Na(+) currents at high temperatures by SCN9A mutations underlies the episodic heat-enhanced pain in inherited erythromelalgia. Sci Rep 2019, 9, 12251, doi:10.1038/s41598-019-48672-6.
  3. Huang, C.W.; Lai, H.J.; Lin, P.C.; Lee, M.J. Changes of Resurgent Na(+) Currents in the Nav1.4 Channel Resulting from an SCN4A Mutation Contributing to Sodium Channel Myotonia. Int J Mol Sci 2020, 21, doi:10.3390/ijms21072593.
  4. Huang, C.W.; Lai, H.J.; Lin, P.C.; Lee, M.J. Changes in Resurgent Sodium Current Contribute to the Hyperexcitability of Muscles in Patients with Paramyotonia Congenita. Biomedicines 2021, 9, doi:10.3390/biomedicines9010051.
  5. Huang, C.W.; Lin, P.C.; Chen, J.L.; Lee, M.J. Cannabidiol Selectively Binds to the Voltage-Gated Sodium Channel Nav1.4 in Its Slow-Inactivated State and Inhibits Sodium Current. Biomedicines 2021, 9, doi:10.3390/biomedicines9091141.
  6. Farinato, A.; Altamura, C.; Imbrici, P.; Maggi, L.; Bernasconi, P.; Mantegazza, R.; Pasquali, L.; Siciliano, G.; Lo Monaco, M.; Vial, C., et al. Pharmacogenetics of myotonic hNav1.4 sodium channel variants situated near the fast inactivation gate. Pharmacological research 2019, 141, 224-235, doi:10.1016/j.phrs.2019.01.004.
  7. Morgan, K.; Stevens, E.B.; Shah, B.; Cox, P.J.; Dixon, A.K.; Lee, K.; Pinnock, R.D.; Hughes, J.; Richardson, P.J.; Mizuguchi, K., et al. beta 3: an additional auxiliary subunit of the voltage-sensitive sodium channel that modulates channel gating with distinct kinetics. Proc Natl Acad Sci U S A 2000, 97, 2308-2313, doi:10.1073/pnas.030362197.
  8. Kazen-Gillespie, K.A.; Ragsdale, D.S.; D'Andrea, M.R.; Mattei, L.N.; Rogers, K.E.; Isom, L.L. Cloning, localization, and functional expression of sodium channel beta1A subunits. J Biol Chem 2000, 275, 1079-1088.
  9. Yu, F.H.; Westenbroek, R.E.; Silos-Santiago, I.; McCormick, K.A.; Lawson, D.; Ge, P.; Ferriera, H.; Lilly, J.; DiStefano, P.S.; Catterall, W.A., et al. Sodium channel beta4, a new disulfide-linked auxiliary subunit with similarity to beta2. J Neurosci 2003, 23, 7577-7585.
  10. Qu, Y.; Isom, L.L.; Westenbroek, R.E.; Rogers, J.C.; Tanada, T.N.; McCormick, K.A.; Scheuer, T.; Catterall, W.A. Modulation of cardiac Na+ channel expression in Xenopus oocytes by beta 1 subunits. J Biol Chem 1995, 270, 25696-25701.
  11. Isom, L.L.; De Jongh, K.S.; Patton, D.E.; Reber, B.F.; Offord, J.; Charbonneau, H.; Walsh, K.; Goldin, A.L.; Catterall, W.A. Primary structure and functional expression of the beta 1 subunit of the rat brain sodium channel. Science 1992, 256, 839-842.
  12. Black, J.A.; Dib-Hajj, S.; McNabola, K.; Jeste, S.; Rizzo, M.A.; Kocsis, J.D.; Waxman, S.G. Spinal sensory neurons express multiple sodium channel alpha-subunit mRNAs. Brain Res Mol Brain Res 1996, 43, 117-131.
  13. Jarecki, B.W.; Piekarz, A.D.; Jackson, J.O., 2nd; Cummins, T.R. Human voltage-gated sodium channel mutations that cause inherited neuronal and muscle channelopathies increase resurgent sodium currents. J Clin Invest 2010, 120, 369-378, doi:10.1172/JCI40801.

Reviewer 3 Report

The paper proposed by Lee and colleagues presents the biophysical properties of SCN4A voltage-gated sodium channels with missense mutations reported on patients with paramyotonia congenita. Data were obtained from patch-clamp experiments after gene expression on cell lines. Current properties were extensively characterized in the whole cell configuration. The results reveal that the two mutants exhibit modifications in the activation as well as inactivation process of the channel leading to an increase in window current. It also appears a modification in the resurgent current for both mutants. Altogether the study indicates that mutations lead to channel dysfunction which may be implicated in the development of the pathology.

The electrophysiological characterization of the mutants is extensively done with high quality and provides convincing results. Figures are well prepared and present appropriately the results. The bibliography is up to date.

I have several remarks to improve the paper:

1- In the method section, the authors indicate that CHO-K1 as well as HEK-293T cells were used for transfection and patch-clamp measurements. Firstly, the authors do not indicate why two different cell lines were used. Secondly, there is no indication, all along the results, of the cell line which was used for each specific experiment. Did the two mutants were transfected in the same cell line ? Was also the WT transfected in the same cell line ? This has to be specified precisely to exclude any artifact in the results due to the use of different cell lines for the same experiment.

2- In the methods section, the authors indicate that cells were treated with protease before patching. Please indicate the rationale of this process. Was it done to separate the cells so that they can be moved into the recording dish with the patch pipette?

3- In the discussion, line 472, the authors indicate a model with two open states. To facilitate the comprehension of this model, it could be helpful to add a new figure with its representation.

Author Response

Response to Reviewer#3 Comments

We thank the reviewer for the positive appraisal and very helpful advice on the manuscript. We have carefully revised the manuscript.

In the method section, the authors indicate that CHO-K1 as well as HEK-293T cells were used for transfection and patch-clamp measurements. Firstly, the authors do not indicate why two different cell lines were used. Secondly, there is no indication, all along the results, of the cell line which was used for each specific experiment. Did the two mutants were transfected in the same cell line ? Was also the WT transfected in the same cell line ? This has to be specified precisely to exclude any artifact in the results due to the use of different cell lines for the same experiment.

RESPONSE:

To avoid the false positive or negative results due to the cell-type specific effects, we carried out the experiments in two different cell lines. The results from the two types of cells line (CHO-K1 and HEK-293T) are similar. Thus, we present the results from what we obtained from CHO-K1 cells.

In the methods section, the authors indicate that cells were treated with protease before patching. Please indicate the rationale of this process. Was it done to separate the cells so that they can be moved into the recording dish with the patch pipette?

RESPONSE:

Protease XXIII is an enzyme used to break down proteins by hydrolyzing peptide bonds. It can catabolize proteins by hydrolysis of peptide bonds. Proteases are inactivated by serine active-site inhibitors, such as phenylmethyl sulfonyl fluoride (PMSF) and diisopropylfluorophosphate. We used it to separate the cells so that they can be placed in the recording dish and made it easy to be attached by the patch pipette [1-5].

In the discussion, line 472, the authors indicate a model with two open states. To facilitate the comprehension of this model, it could be helpful to add a new figure with its representation.

RESPONSE:

Thanks for the reviewer`s comment. We have reported the model with a depicted figure already [1,2]. Since there are robust evidences in our previous studies on Nav1.7 and Nav1.4 channels [1-5], we have the strong argument that there probably are two open states for these voltage sodium channels which generate both transient and resurgent sodium currents. However, to avoid any duplication and the suspected self-plagiarism, we only explain by text with references.

Reference

  1. Huang, C.W.; Lai, H.J.; Huang, P.Y.; Lee, M.J.; Kuo, C.C. The Biophysical Basis Underlying Gating Changes in the p.V1316A Mutant Nav1.7 Channel and the Molecular Pathogenesis of Inherited Erythromelalgia. PLoS biology 2016, 14, e1002561, doi:10.1371/journal.pbio.1002561.
  2. Huang, C.W.; Lai, H.J.; Huang, P.Y.; Lee, M.J.; Kuo, C.C. Anomalous enhancement of resurgent Na(+) currents at high temperatures by SCN9A mutations underlies the episodic heat-enhanced pain in inherited erythromelalgia. Sci Rep 2019, 9, 12251, doi:10.1038/s41598-019-48672-6.
  3. Huang, C.W.; Lai, H.J.; Lin, P.C.; Lee, M.J. Changes of Resurgent Na(+) Currents in the Nav1.4 Channel Resulting from an SCN4A Mutation Contributing to Sodium Channel Myotonia. Int J Mol Sci 2020, 21, doi:10.3390/ijms21072593.
  4. Huang, C.W.; Lai, H.J.; Lin, P.C.; Lee, M.J. Changes in Resurgent Sodium Current Contribute to the Hyperexcitability of Muscles in Patients with Paramyotonia Congenita. Biomedicines 2021, 9, doi:10.3390/biomedicines9010051.
  5. Huang, C.W.; Lin, P.C.; Chen, J.L.; Lee, M.J. Cannabidiol Selectively Binds to the Voltage-Gated Sodium Channel Nav1.4 in Its Slow-Inactivated State and Inhibits Sodium Current. Biomedicines 2021, 9, doi:10.3390/biomedicines9091141.

Round 2

Reviewer 1 Report

see attached

Author Response

Response to Reviewer#1 Comments

We have carefully revised the manuscript with responses one-by-one successively as follows.

[1] In line 180, the Boltzmann equation needs to be revised. Please change to the parameters of “gating charge in q”. If possible, please estimate changes in free energy based on the Boltzmann equation.

RESPONSE:

According to the reviewer's suggestions, we have revised the parameters of the “gating charge in q” of the Boltzmann equation in the revised manuscript [lines 179, 180, 187, and 192].

When the open (O) and close (C) states of a voltage-dependent ion channel reaches equilibrium.

C↔O

And the free energy change between these two states has been expressed as, ΔG. Then, the ratio of these two states (O/C) can be expressed as the following equation.

Where kB is the Boltzmann constant and T is the absolute temperature.

According to the semilogarithmic plot to predict the fraction of open channels with different charge, the higher the charge, the steeper the rising part of the curve (the following figure, from Hille Bertil; Ion channels of Excitable Membrane; page 57-59). These curves can be related to the actual voltage dependence of peak gNa and gk.

In this simple model with following Hodgkin and Huxley’s treatment, a lower limit for the magnitude of the gating charge per channel can be calculated from the steepness of the voltage dependence of gaiting. As we proposed the transition of C to O is a conformational change that moves a gating charge of valence zg from the inner to the outer membrane surface, across the membrane potential drop E. Thus, there will be two energy change of transition (C to O). In the absence of a membrane potential (E =0), the conformational energy increase upon opening the channel is set to be w. The other term is the voltage-dependent one due to movement of the gating charge zg when there is a membrane potential. This electrical energy increase is -zgqeE, where qe is the elementary charge, and the total energy change becomes (w – zgqeE). In terms of energy change and gives the voltage dependence of gating in the system, the Boltzmann equation dictates the ratio of open to closed channels at equilibrium is

After rearranging, the fraction of open channels can be represented as the following formula,

where the zgqeE can be expressed as zeV (V, is the voltage across the membrane).

Thus, it can be rearranged into,

Therefore,

DG = w – zeV

Then, the ratio of the number of channels in the open state to all channels which equals to the ratio of the recorded current I to the maximum current Imax, can be presented a the following equation:

In terms of current ratio (I/Imax), the Boltzmann function used to fit our experiment findings (subtitle 2.5 in the Method section) is

Therefore, 

Since ΔG = wzeV, the equation above can be rewritten as,

The ΔG can then be calculated by the formula,

DG = w – zeV = ( )/kslope * kB * T

Assuming the voltage to be 0 at baseline (V = 0), the change in free energy (ΔG) in the WT, p.V781I and p.A1737T channels can be obtained and listed as the following table. 

channel

Free energy change (ΔG)

WT Nav1.4

3.9

p.V781I

4.7

p.A1737T

3.8

However, this article focuses on investigating the pathophysiology of the changes in gating and resurgent currents in paramyotonia congenita. Such details in physics is beyond the scope of this study.

[2] In line 259 and 358, the k value is virtually distinguishable from the gating charge (q), since beta peptide could be a gating modifier. Please change the formula of Boltzmann equation in the manuscript. Figures 2D and 5D needs to be corrected.

RESPONSE:

According to the reviewer's suggestions, we have revised the parameters of the “gating charge in q” in lines 231 259, and 358 of the Boltzmann equation. Figures 1D, 2D, and 5D corrected in the revised manuscript.

[3] In lines 482-483, the sentence needs to be rephrased.

RESPONSE:

As suggested, we have repheased the sentence to “There might be a change in binding energy when interacting with the internal Navb4 peptides “ [line 481-482].

[4] The text shown in lines 520-534 appears to be irrelevant to the present study. The results quoted are primarily present in native cells, such as dorsal root ganglion neurons. The main point is that the Na+ currents demonstrated in the work were shown in CHOK-K1 or HEK293T cells. How can the insertion of beta subunit(s) (lines 168-170) “functionally”, “stochiometrically” and “allosterically” associate the overexpressed Nav a-subunits in CHO-K1 or HEK293T cells? (Hence, in lines 484-485, the statement tends to be speculative). It is possible unless a and b subunits of Nav channels could be overexpressed with the same cells.

RESPONSE:

While considering the irrelevant to the main findings in the study, the sentences in lines 518-533 have been left out in the revised manuscript. Several studies have investigated the resurgent current using the over-expressed a-subunits and the b4 peptide using CHO-K1 or HEK293T cells [1-11]. In previous study, knockout mice to ablation the expression of Navβ4 cannot generate the resurgent current [12,13]. Our previous study found that the resurgent current cannot be detected unless the adding of peptide of Navβ4 [7,9].

In this, Nav1.4 and our previous studies in Nav1.7 channels, a few similar findings show that there are different activation and decay kinetics for resurgent and transient currents. The gating charge (q, the slope) for these two currents are quite different. The time constant for decay in resurgent is also different from the transient current. We proposed that there might be at least two different states for transient and resurgent currents. Although it might be speculative, the β4 peptide may play a role as a modifier protein mediating the open state for resurgent current at repolarizing stage.

[5] Some of grammatical errors need to be modified and rephrased.

RESPONSE:

In the revised manuscript, the manuscript has been revised to correct the errors. The manuscript has been reviewed by a native English speaker to improve readability.

Reference

  1. Theile, J.W.; Jarecki, B.W.; Piekarz, A.D.; Cummins, T.R. Nav1.7 mutations associated with paroxysmal extreme pain disorder, but not erythromelalgia, enhance Navbeta4 peptide-mediated resurgent sodium currents. J Physiol 2011, 589, 597-608, doi:10.1113/jphysiol.2010.200915.
  2. Theile, J.W.; Cummins, T.R. Inhibition of Navbeta4 peptide-mediated resurgent sodium currents in Nav1.7 channels by carbamazepine, riluzole, and anandamide. Mol Pharmacol 2011, 80, 724-734, doi:10.1124/mol.111.072751.
  3. Patel, R.R.; Barbosa, C.; Xiao, Y.; Cummins, T.R. Human Nav1.6 Channels Generate Larger Resurgent Currents than Human Nav1.1 Channels, but the Navbeta4 Peptide Does Not Protect Either Isoform from Use-Dependent Reduction. PLoS One 2015, 10, e0133485, doi:10.1371/journal.pone.0133485.
  4. Mason, E.R.; Cummins, T.R. Differential Inhibition of Human Nav1.2 Resurgent and Persistent Sodium Currents by Cannabidiol and GS967. Int J Mol Sci 2020, 21, doi:10.3390/ijms21072454.
  5. Ahn, H.S.; Dib-Hajj, S.D.; Cox, J.J.; Tyrrell, L.; Elmslie, F.V.; Clarke, A.A.; Drenth, J.P.; Woods, C.G.; Waxman, S.G. A new Nav1.7 sodium channel mutation I234T in a child with severe pain. Eur J Pain 2010, 14, 944-950, doi:10.1016/j.ejpain.2010.03.007.
  6. Cummins, T.R.; Dib-Hajj, S.D.; Waxman, S.G. Electrophysiological properties of mutant Nav1.7 sodium channels in a painful inherited neuropathy. J Neurosci 2004, 24, 8232-8236, doi:10.1523/JNEUROSCI.2695-04.2004.
  7. Huang, C.W.; Lai, H.J.; Huang, P.Y.; Lee, M.J.; Kuo, C.C. The Biophysical Basis Underlying Gating Changes in the p.V1316A Mutant Nav1.7 Channel and the Molecular Pathogenesis of Inherited Erythromelalgia. PLoS biology 2016, 14, e1002561, doi:10.1371/journal.pbio.1002561.
  8. Huang, C.W.; Lai, H.J.; Huang, P.Y.; Lee, M.J.; Kuo, C.C. Anomalous enhancement of resurgent Na(+) currents at high temperatures by SCN9A mutations underlies the episodic heat-enhanced pain in inherited erythromelalgia. Sci Rep 2019, 9, 12251, doi:10.1038/s41598-019-48672-6.
  9. Huang, C.W.; Lai, H.J.; Lin, P.C.; Lee, M.J. Changes of Resurgent Na(+) Currents in the Nav1.4 Channel Resulting from an SCN4A Mutation Contributing to Sodium Channel Myotonia. Int J Mol Sci 2020, 21, doi:10.3390/ijms21072593.
  10. Huang, C.W.; Lai, H.J.; Lin, P.C.; Lee, M.J. Changes in Resurgent Sodium Current Contribute to the Hyperexcitability of Muscles in Patients with Paramyotonia Congenita. Biomedicines 2021, 9, doi:10.3390/biomedicines9010051.
  11. Lewis, A.H.; Raman, I.M. Interactions among DIV voltage-sensor movement, fast inactivation, and resurgent Na current induced by the NaVbeta4 open-channel blocking peptide. J Gen Physiol 2013, 142, 191-206, doi:10.1085/jgp.201310984.
  12. Grieco, T.M.; Malhotra, J.D.; Chen, C.; Isom, L.L.; Raman, I.M. Open-channel block by the cytoplasmic tail of sodium channel beta4 as a mechanism for resurgent sodium current. Neuron 2005, 45, 233-244, doi:10.1016/j.neuron.2004.12.035.
  13. Aman, T.K.; Grieco-Calub, T.M.; Chen, C.; Rusconi, R.; Slat, E.A.; Isom, L.L.; Raman, I.M. Regulation of persistent Na current by interactions between beta subunits of voltage-gated Na channels. J Neurosci 2009, 29, 2027-2042, doi:10.1523/JNEUROSCI.4531-08.2009.

Reviewer 2 Report

Lee and collaborators have revised the manuscript and responded to a number of my comments. However, there are still a number of issues requiring revision. English language should be carefully revised.

Author Response

Response to Reviewer#2 Comments

Lee and collaborators have revised the manuscript and responded to a number of my comments. However, there are still a number of issues requiring revision. English language should be carefully revised.

RESPONSE:

Many thanks for the comments. This manuscript has been reviewed by a native English speaker to improve readability. Certification is presented as the following sheet. In addition, the manuscript has been revised by another scholar with improvement in English writing.

Round 3

Reviewer 1 Report

There are serious errors shown in the following:

[1] In lines 178-180, line 187, lines 230-231, line 257, the equation is virtually incorrect. “q” definitely is not equivalent to “k” (slope factor).

[2] Since “q” and “k” are distinguishable, the value of free energy could also be incorrect. By the way, please show the unit of free energy change (J/mol or kcal/mol?) shown in the reply of comments.

Author Response

Response to Reviewer#1 Comments (round 3)

We have carefully revised the manuscript with the responses one-by-one to the questions successively as follows.

[1] In lines 178-180, line 187, lines 230-231, line 257, the equation is virtually incorrect. “q” definitely is not equivalent to “k” (slope factor).

RESPONSE:

Thanks reviewer’s helpful suggestions. We corrected the sentences and replaced “q” by “k” (slope factor) in the lines 179-180, line 187, lines 230-231, and line 257 in the revised manuscript.

[2] Since “q” and “k” are distinguishable, the value of free energy could also be incorrect. By the way, please show the unit of free energy change (J/mol or kcal/mol?) shown in the reply of comments.

RESPONSE:

According to the reviewer’s comment, kT (also written as kBT) is the product of the Boltzmann constant, k (or kB), and the absolute temperature, T. This product is used in physics as a factor for energy values in molecular-scale (sometimes it is used as a unit of energy). The free energy is one of the determinant factors for molecular dynamics and distribution; nevertheless, the ratio of free energy and kT (E/kT) (see Arrhenius equation, Boltzmann factor) may be even more critical for the equilibrium of molecular distribution across the membrane. A “canonical ensemble” is the statistical ensemble that the possible states of a mechanical system in thermal equilibrium with a heat bath at a fixed temperature. The system can exchange energy with the heat bath, so that the states of the system will differ in total energy. For a system in equilibrium in canonical ensemble, the probability of the system in state with energy E is proportional to e−ΔE/kT. The “kT” is the amount of heat required to increase the thermodynamic entropy of a system by k.

Approximate values of kT at 298  K

Units

kT = 4.11×1021

J

kT = 4.114

pN⋅nm

kT = 9.83×10−22

cal

kT = 25.7

meV

kT=-174

dBm/Hz

Assuming the voltage to be 0 at baseline (V = 0), the change in free energy (ΔG) in the WT, p.V781I and p.A1737T Nav1.4 channels can be obtained and listed as the following table. 

Nav1.4 channel

Free energy change (ΔG)

WT

1.602910-20  J

p.V781I

1.931710-20  J

p.A1737T

1.561810-20  J

 Reference: Atkins' Physical Chemistry, 9th ed., by P. Atkins and J. dePaula, Oxford University Press
